# Crop Diversity in Agroecosystems for Pest Management and Food Production

**DOI:** 10.3390/plants13081164

**Published:** 2024-04-22

**Authors:** Jillian Lenné, David Wood

**Affiliations:** North Oldmoss Croft, Fyvie, Tirriff AB53 8NA, UK; agrobiodiversity@gmail.com

**Keywords:** monodominance, monoculture, genetic resistance response, escape response, crop introduction, field margins, irrigated rice systems

## Abstract

During the past 30 years, there has been a growing belief in and promotion of agroecosystem diversity for pest management and future food production as an agroecological or nature-based approach. Monoculture agriculture, which produces most of our food, is considered to be highly vulnerable to pests in contrast to plant species-diverse agroecosystems which may possess a greater abundance of natural enemies, keeping pest populations under control. In this paper, we question the role of crop diversity for pest management and explore the relationship between crop and associated diversity and pests through the following processes: environmental stresses that favor monodominance; evolutionary adaptations that resist insect herbivores (genetic resistance response); mechanisms of spatial escape from insect herbivores (escape response); and the role of crop-associated biodiversity. We present strong evidence that not only questions the high vulnerability of monocultures to pest damage but also supports why monocultures continue to produce most of the world’s food. Reference is made to the importance of targeted plant breeding and the role of trans-continental crop introduction supported by efficient quarantine for pest management. We conclude that—with the exception of irrigated rice—much more research is needed to better understand the role of crop diversity in agroecosystems for pest management and food production.

## 1. Context

The main focus of this perspective paper is on the different ways that an understanding of natural ecology underpins crop production for pest management [1]. There has been increasing emphasis in the past 30 years on the need to diversify agroecosystems based on the premise that such diversity is an important source of predators and parasitoids of crop pests and, as a result, pests will be managed naturally [2,3,4,5]. This is further supported by the premise that agricultural monocultures, which produce most of our food, are highly vulnerable to pests [6]. But is this true? The role of diversity was already questioned almost 30 years ago by Reid [7], who argued that ecologists had long ago discarded the idea of a positive relationship between biotic diversity and stability. However, Reid considered that the notion was probably too useful for environmentalists ever to reject. Reid’s observation seems still to apply to those who consider natural monodominant vegetation to be an unsuitable model for monoculture fields.

In this paper, we will heed Reid’s warning and further explore the relationship between plant diversity and pests in an attempt to make crop production more nature-based without relying on the solution of more species diversity.

There are five goals of this perspective study:To summarize some of the arguments promoting species diversity in fields for stability under biotic stress;To highlight the existence of natural plant monodominance as an adaptation to abiotic stresses;To present arguments for the maintenance of monodominance through adaptation to biotic stress (evolved resistance, augmented by targeted crop improvement for pest resistance);To explain a major form of plant monodominance after separation from co-evolved pests and diseases (escape response obtained by trans-continental crop introduction supported by quarantine); andTo explore the role of crop-associated biodiversity in managing pests with emphasis on field margins and monoculture irrigated rice agroecosystems.

## 2. Definitions

To ensure clarity and avoid different interpretations and misunderstandings in this perspective paper, Box 1 defines the common terminology used.

Box 1Definitions of commonly used terminology.**Monodominance** (https://en.wikipedia.org/wiki/Monodominance, accessed on 29 March 2024): originally coined for forest trees, it is a plant stand where most plants (60% or more) are of the same species; this does not preclude intra-specific varia-tion.
**Monoculture** (https://languages.oup.com/google-dictionary-en/, accessed on 29 March 2024): the cultivation of a single crop in a given area—as small, medium or large fields.
**Agroecosystem** (https://biodiversity.europa.eu/europes-biodiversity/ecosystems/agroecosystems, accessed on 29 March 2024): communities of plants and animals interacting in the environment that has been modified by people to produce food, fiber, fuel, and other products for human consumption and processing.
**Crop diversity** (https://en.wikipedia.org/wiki/Crop_diversity, accessed on 29 March 2024): the variety and variability of crop species used in agriculture, including their genetic and phenotypic characteristics; the emphasis in this paper is on field diver-sity.
**Genetic resistance:** used against pests in the context of natural plant defenses against insect pests and crop resistance through breeding.
**Trans-continental crop introduction** (https://en.wikipedia.org/wiki/Columbian_exchange, accessed on 29 March 2024): introduction of new staple food crops from one continent into another, as exempli-fied by the Columbian exchange.Escape against pests: used in the context of moving crops away from centers of di-versity to escape pests, such as through trans-continental crop introduction.
**Crop-associated biodiversity** (https://www.fao.org/agriculture/crops/thematic-sitemap/theme/biodiversity/cab/vn/, accessed on 29 March 2024): this includes plant and non-plant diversity in and around fields where the crop is being grown, e.g., bunds, hedgerows, and field mar-gins.

Some of the terminology used in this perspective paper has been the subject of widespread criticism both in the literature and media (https://www.bloomsbury.com/uk/monocultures-of-the-mind-9781856492188/, https://www.butterfliesandwheels.org/2003/the-anti-monoculture-mania/, accessed on 29 March 2024). It is unfortunate that the term “monoculture” is generally viewed only in the modern sense of “industrial agriculture”, and as Sumberg and Giller [8] noted, the term has been weaponized. Monoculture agriculture is the most globally widespread form of agriculture practiced by small, medium and large farms using a wide range of management methods for millennia and produces most of our food [9,10]. As defined above, a monoculture is a single crop—a monospecific stand—in a given area which may be small or large. A monoculture can be made up of a single crop variety (many of which are functionally diverse), a mixture of varieties of the same species or a traditional landrace mixture. The definition of monoculture does not include how it is managed.

We acknowledge the important role of inter-crops in some cropping systems, especially cereal–legume systems, and this is used as an example in the section on crop-associated biodiversity. The term “pest” refers to insect pests and diseases. We acknowledge the importance of weeds in agricultural systems but this paper is focused on pests.

## 3. The Promotion of Species Diversity in Fields for Stability under Biotic Stress

Conway [11] highlighted the importance of maintaining and, wherever possible, enhancing diversity, as more diverse agroecosystems tend to be more sustainable and, often, more productive than systems which are otherwise comparable. There are other such claims, some reviewed by Wood [12]. For example, Altieri [13] thought that a major critical concept (of agroecology) was that modern agroecosystems resembled immature simplified ecosystems and therefore lacked the ability to regulate pest populations. Similarly, Altieri and Nicholls [4] suggested that agriculture implied the simplification of the structure of the environment over vast areas by replacing nature’s diversity with a small number of cultivated plants and domesticated animals. However, natural monodominant vegetation can be found under a range of ecological conditions of stress (see Section 4). And there is one type of stress—substrate disturbance—that crop fields still mimic through ploughing and tilling in support of crop monocultures. In contrast, the belief persists that little is “natural” in contemporary agriculture. For example, it was claimed that it was not nature’s way to allow large expanses of land to be planted for a single crop [14] (p. 42). This view is challenged by the widespread existence of present-day cereal fields of “contemporary agriculture”, which are close mimics of the naturally monodominant vegetation of their wild relatives [15,16]. Moreover, Van der Werf and Bianchi [17] addressed the need to “learn from nature” in order to reduce pesticide use in cropping; they concluded that diversification is not a silver bullet.

## 4. Existence of Stable Plant Monodominance under Abiotic Stress

Abiotic stress appears to be an ecological factor that favors monodominance. It seems that stress selects for adapted species; the higher the level of stress, the lower the number of adapted species and the more monodominant the vegetation. Finally, at the highest levels of stress, no species is sufficiently adapted to survive. Monodominance is important in the origin of some field crops, e.g., the founder cereals—wheat, barley, and rice. Indeed, we have suggested—with numerous examples—that certain types of natural monodominance under conditions of strongly wet–dry seasonality were of major relevance to crop domestication [10]. Two types of ecological stress were considered (both leading to monodominant vegetation where only one, tough, species can survive):Seasonally naturally flooded wetland vegetation, eventually leading to floodwater farming, of special importance for rice; andThe probable role of natural grassland fires in the ecology of wild relatives of the first cereals—wheat and barley—in and around the Fertile Crescent.

To become monodominant and out-compete other species, plant species must have evolutionary adaptations to abiotic stresses. The most comprehensive example is for the closest wild relatives of the first cereals [16]. A novel explanation of the origin of cereal agriculture was proposed, based on the ecology and adaptive morphology of wild cereals ancestral to our founder cereals (einkorn, emmer, and barley). Wild cereals are unusually large seeded. A natural evolutionary–ecological syndrome involves large seeds, awns, and monodominance (LAM). Awns bury attached seeds in the soil, protecting the seeds from fire; buried seeds need to be large to emerge on germination; large seeds, growing without competition from small-seeded plants, will produce monodominant vegetation (which exists in regions of cereal domestication). Climatic and edaphic instability at the Pleistocene–Holocene boundary would have provided an impetus for the spread of annual ruderal grasses. LAM grassland provided an obvious natural model for the origins of cereal agriculture. The fact that monodominance is a long-standing characteristic of the natural LAM syndrome validates cereal monocultures (now producing much of our staple food). This was further explored by Petersen and Kellogg [18] with respect to the role of awns.

Furthermore, a partial review of evidence showed that many wild grasses related to cereals are monodominants [1]. Examples of monodominance include crop relatives of rice, such as *Oryza barthii* in Africa, *O. longistaminata* in Mali, *Oryza australiensis* and *O. meridionalis* in tropical Australia, *O. longiglumis* in Papua New Guinea, and *O. minuta* in Indonesia. One discovery from older studies was the wild Asian rice relative, *Oryza coarctata*, which grows in vast monodominant stands on regularly flooded coastal mud-flats at the mouth of the Indus River [19]. The adaptations of wild rice and also *Echinochloa stagnina* and *Saccharum spontaneum* probably depend on annual water-depth changes and fire, as suggested for the plant monodominance of the Brazilian Pantanal [20]. Possibly adapted to fire regimes, there is monodominant wild *Sorghum* in Africa and ancestors of the first cereals *Triticum monococcum* subsp. *baeoticum*, *T. turgidum* subsp. *dicoccoides*, *Hordeum vulgare* subsp. *spontaneum*, in the Eastern Mediterranean, and *Avena sterilis* and *Secale montanum* [16]. As this was not a detailed search, there are most likely to be more monodominant relatives of cereals, which further supports the widespread practice of growing staple cereal crops in monocultures. As this perspective mainly focuses on cereals, it would be interesting to explore the ecology of wild relatives of other crop domesticates such as legumes, roots and tubers, and oil seed crops.

It is worth noting that annual wild relatives of cereals can produce each generation—seed to seed—in the wet season, with the buried seed protected from dry-season fire. A general comment can be made: seasonal flooding for rice and seasonal fire for other cereals seem to explain the two sources of ecological stress needed to promote adaptation for monodominance of crops grown in seasonal climates. More research is needed on the role of other abiotic stresses. A caution is needed: flood and fire stress will not inevitably lead to monodominance and then domestication. For example, seasonally flooded and burned Pantanal vegetation in South America has produced no domesticates.

Remarkably, the earliest cereal domestications were of species of several different grass tribes that by convergent evolution had already evolved the ability to grow naturally as monodominants. When subsequently, during the process of domestication, they were grown in fields, they were *preadapted* to monodominance. Almost inevitably, when cultivated in monocultures, these early domesticates had brought with them other features of their adaptation in natural vegetation, that is, stability and persistence. It can be claimed that monodominant fields of the first cereals were close mimics of their previous natural monodominance. However, the previous disturbance of flood and fire had to be replaced in crop fields by bunding and field puddling for rice and ploughing, tilling, and weeding for other cereals.

In these early cereals, close dependence on nature to provide natural models for fields has not prevented a remarkable ignorance of the existence of natural plant natural monodominant vegetation and its value as a model for fields. For example, a key early paper argued that species-diverse vegetation was needed as a protection against herbivorous insects [21], that is, the opposite of the evolved adaptation of species that allowed natural monodominance. However, Janzen soon realized that the idea of the pressure of herbivores on plants was not the only explanation of vegetational structure: there were other possibilities. Soon after, Janzen [22], in a paper on tropical forest diversity, himself suggested (somewhat obliquely) that the “existence of such (dipterocarp) forests … falsified the dogma that diversity is mandatory for ecosystem stability in highly equitable climates”. Unfortunately for crop production, this cautionary statement was ignored.

Crop scientists and others then followed Janzen in producing counter-claims to the general application of the prevailing diversity–stability hypothesis. For example, May [23] noted that there is no reason to expect simple natural monocultures to be unstable, giving as examples bracken (*Pteridium aquilinum*) and *Spartina*. Bazzaz [24] argued that early successional communities are likely to have low species diversity as each species occupies a large portion of total available resources (niche space) and commented that low diversity had been shown for a number of early successional sequences. Grime [25] noted that it would be naive to assume that species-poor ecosystems are always malfunctional; some of the world’s most extensive and ancient ecosystems—boreal forests, bogs, and heathlands—contained few species. In a more extensive comment on the diversity–stability hypothesis, Dover and Talbot [26] observed that if it is true, it provides a strong practical argument for conservation—for maintaining diversity in ecosystems whenever possible as a natural buffer against perturbation. But if the theory is flawed, basing policies on it could have unexpected and undesirable environmental consequences. For instance, if diversity causes stability, the most species-rich communities—such as tropical rain forests and coral reefs—should be able to withstand the greatest disruption at human hands. In fact, these communities are among the most fragile [27,28,29]. Dover and Talbot [26] concluded that experimental evidence and theoretical analysis questioned the belief that diversity causes stability. It was oversimplified.

The paper has already shown that natural monodominance is widespread, under a range of abiotically stressed conditions. A sound ecological paradigm for monodominance had been provided by the ecologist Grime [30], who located plant strategies within a triangle determined by species abilities such as *ruderality*, *stress tolerance*, or *competitiveness*. Importantly, Grime noted that at the respective corners of the triangle, members of each of the three classes become the exclusive constituents of the vegetation. That is, as ecological stress increases selection pressure, members of each class can become monodominant. In effect, they are *tougher* than any other available species under these conditions of stress.

In a colloquium on species diversity, Tilman [31] stressed the vulnerability of monocultures based on ecological principles. However, Janzen pointed out that monocultures occur throughout nature in a wide variety of circumstances and Ruttan noted examples of monocultural systems, such as East Asian wet rice culture, that have been sustained over several centuries [32]. Wood and Lenné [16] suggested that natural monodominance played a key role in the domestication of our first cereals more than 10,000 years ago. Given the monodominance of these crop wild relatives, Tilman’s support for polycultures does not appear to be universally applicable. More generally, if evidence for mixed-species vegetation is true, then the persistence of natural monospecific vegetation should be ecologically impossible, as over time, monodominant vegetation will always be invaded and replaced by multi-species vegetation. However, natural monodominant vegetation exists widely and must have, in order to survive, resistance mechanisms against pests and disease. While this is of obvious advantage to domesticated crops, it is not enough to ensure domestication. Other features are necessary, for example, the large seed and seed burying of the relatively rare LAM syndrome grasses [16]. Even so, grasses can be domesticated without large seeds. Wild sugar cane, the monodominant *Saccharum spontaneum*, was domesticated for its sugary stem.

## 5. Maintenance of Monodominance by Adaptation to Biotic Stresses (Genetic Resistance Response)

But once monodominance has been achieved (by adaptation to abiotic stresses), there is an evolutionary dilemma for the monodominant species: current theory argues that such uniform vegetation will be subject to greater biotic pressures, for example, insect herbivory, than mixed-species vegetation. That is, monodominant vegetation will inevitably become a target for pests and pathogens. For example, Ridley [33] claimed that where too many plants of one species were grown together, they were apt to be attacked by some pest, insect, or fungus. Indeed, this premise is the basis for the calls for polycultures in crop fields, with the associated rejection of monocultures as too fragile. But, as explained above, monodominant vegetation exists widely, so the “fragility” belief is not general (in fact, it leads to the ignoring of *all* monodominant vegetation as unnatural artefacts).

Rather than ignoring monodominance, it is possible and useful to try to explain it. Any explanation must conclude that monodominant species are apparently better adapted to survive than other species in the region. For example, Janzen [34], with respect to tropical forest monodominance, asked the following question: “Why don’t those herbivores that can deal with the defences of monoculture tree stands move into these habitats and literally mow them to the ground?” Janzen then suggested that one of the traits for living in a habitat supporting a pure stand should be the evolution of those kinds of chemical and behavioral defenses that are so effective that the plant does not rely on spatial escape. Janzen went on to suggest promising new areas of research: one of these was to “figure out how tropical plants survive in pure stands, rather than worry about the mixed species stands”. Those promoting species diversity for crop stability have tended to ignore this “figuring out”. Clayton and Renvoize [35] (pp. 16–17) did figure it out for grasses: “grasses benefit from a fire regime that is lethal to many other plants, and, having co-evolved with herbivores, can sustain a level of predation sufficient to cripple many competitors”. It is the ability of grasses to “cripple competitors” that allows natural monodominance.

If species monodominance increases insect pressure (generally accepted), then it, inevitably, will also tend to increase an associated adaptation of plant species to resist insects. Janzen [34] suggested “chemical and behavioural defences”, notably by plants evolving toxicity. Coley and Barone [36] proposed that insect herbivores, which typically have a narrow host range in the tropics, mainly feed on leaves and have selected for a wide variety of chemical, developmental, and phenological defenses in plants. For example, a wide range of plant species evolve anti-feeding compounds, such as resin, latex, tannins, and very many insecticidal toxins. Some (but not all) of these are used by monodominant plants, for example, *Rhizophora* mangroves, rich in tannin. Knowledge of the existence of anti-feeding compounds has generated increased exploration of gene pools of monoculture crops for resistance traits, which include both chemical compounds, such as proteinases, and physical barriers, such as waxy plant surfaces and glandular trichomes. This research supports the continuing role of targeted crop breeding—both conventional and biotechnological—with the purpose of generating improved varieties of many important food crops with resistance to major pests [37,38,39,40].

## 6. Maintenance of Monodominance by Removal from Co-Evolved Pests and Disease through Plant Introduction (Escape Response)

The positive impact on crop production following introduction to another continent depends on the *asymmetry* between biotic stress and abiotic stress. *Biotic stress* is a result of the presence of co-evolved pests and diseases in the endemic range of the plant. Movement away from these locations of specific constraints allows the plant to invest in production, rather than fight biotic stress, which inevitably lowers yield. In contrast to the local specificity of biotic constraints, *abiotic stress* impacts the species over a much wider geographic range. For example, the term Mediterranean zone (Köppen climate classification Csa) is determined by warm wet winters and hot dry summers and located 30° to 45° north and south of the Equator. This includes Cape Province in South Africa, Western Australia, and some of California and Chile, all allowing wine grape production and crops such as *Citrus* and olive, among many others. The Mediterranean region, as a crop center of origin, has a series of co-evolved biotic constraints for each and every crop, while growing crops in other Mediterranean zones can avoid these constraints [16].

The human introduction of crops to other continents, starting with the land movement of crops across Eurasia in the third millennium BC [41] and, notably, following the discovery of the New World in the year 1492 [42], became trans-oceanic. And, more importantly, its general importance is under-appreciated [15,43]. Apart from rice, most crop production globally is in regions away from the region of domestication (where the highest concentration of co-evolved pests and diseases is found). This phenomenon is well known [44,45,46]. Advantages have also been described clearly by Janzen [47]: “For some tropical crop plants, growing them in new geographic areas and thereby leaving their pests behind is promising, as evidenced by the success of rubber, cacao, cotton, coffee, bananas, and sugar cane, when grown as introduced plants”. As crop introduction is of massive global value, it deserves much more attention.

With the exception of rice in Asia, most food from staple crops comes from their harvests from continents other than the continent of origin; for example, food production in a number of African countries is more than 70% from introduced crops [15,43]. Crop introduction also has a wide application for forest plantation trees, crop tree plantations, and arable crops.

The success of crop introduction depends on the fact that it allows plant species to escape from their co-evolved pests and diseases, although this is not always failsafe (see below). Unfortunately, a few past examples of pathogens introduced many years after initial crop introduction (see Ref. [10] for the history of the 1840s European potato blight) are commonly used to criticize monocultures [6]. Extensive international plant quarantine systems and regulations now largely protect introduced crops.

Perhaps soybean is the most dramatic example of crop introduction. Originating in Southeast Asia, soybean is now principally produced in the Americas as a monoculture, with Brazil leading with 155 million tons (2013), then the USA with 113 million tons and Argentine with 50 million tons. Next is China—a region of domestication of soybean—which only produces 21 million tons [48]. A recent report documents the South American exports related to US exports [49]. It is tempting to relate the increasing dominance of South America in soybean production and export to the promotion of increased crop diversity for Latin America (but not for other regions) in the International Assessment of Agricultural Science and Technology for Development (IAASTD) regional reports [50].

The benefits of escape from pests also extend beyond crops to pasture species such as the grass *Brachiaria brizantha*, introduced from Africa into Brazil, where it occupies, as a monodominant, around 50 million ha. Also introduced from Africa were *Brachiaria decumbens*, *B. humidicola*, and *Panicum maximum*, which are the main pasture grasses used in Brazil. Brazil has been the world’s largest exporter of beef since 2004 [51]. Similarly, for forest plantations, *Pinus radiata*—the Monterey Pine—has become the most widespread monodominant plantation timber tree in the world, from its origin in a small area of California and two small Mexican islands in the Pacific [52] to being supported in many countries by rigorous quarantine systems to avoid risks of introduced pests.

However, a level of caution is needed: even at a distance, species will not be entirely free of pests and pathogens; they may meet with what are called “new-encounter” pests and diseases, to which, of course, they will have evolved no natural resistance [53,54]. In addition to new-encounter problems, there is the added complexity of re-encounters: the accidental introduction of a former problem, by-passing quarantine, may “catch-up” and “re-encounter” the crop. For example, after the accidental introduction of the Cassava mealybug (*Phenacoccus manihoti*), it became a problematic “catch-up” pest of the very important cassava food crop previously introduced to West Africa from South America. In order to control it, parasites of the mealybug were searched for in South America by entomologists from the International Center for Tropical Agriculture (CIAT) in Colombia. A hymenopteran mealybug parasitoid (*Apoanagyrus lopezi*) was discovered in Brazil. This was then distributed widely in the field across 26 African countries. Neuenschwander [55] pointed out that all countries of central and southern Africa in which *A. lopezi* had been released reported good biological control with vastly reduced population levels of *P. manihoti*, and the mealybug was relegated to minor pest status. This was a good example of “real” ecology contributing to food stability, also involving the International Institute of Biological Control CABI (IIBC) at Silwood Park, England. It is worth noting that this, once established, was an effective, pesticide-free, natural biocontrol.

## 7. The Role of Crop-Associated Biodiversity in Managing Pests

A comprehensive analysis of the management of above-ground crop-associated biodiversity (C-AB) for pest management, with emphasis on natural and augmented biological control of pests, can be found in Lenné [56]. This section looks more selectively at the role of C-AB around fields and in the farm landscape to manage crop pests. Special emphasis is given to the irrigated rice agroecosystem which, unlike other important food crop systems, has been the subject of considerable research over more than 40 years.

A number of comprehensive studies in different cropping systems on different continents have concluded that the management of pests through diversity was inconsistent. In a review of diversity for pest management in agroecosystems, Ratnadass et al. [57] observed that it is not necessarily true that vegetational diversification reduces the incidence of pests due to the ability of some pests to use a wide range of plants as alternative hosts/reservoirs. In a meta-analysis of crop pests and natural enemy response to landscape complexity, Chaplin-Kramer et al. [58] noted that despite positive effects on natural enemies, consistent suppression of pests has not been detected in complex landscapes, at least over the timescales of their ecological studies. Wyckhuys et al. [59] looked at the status and potential of conservation biological control (CBC) for 53 crops from 390 literature records. Most were for rice, maize, and cotton. No records were found for yams, taro, sago, breadfruit, papaya, pineapple, avocado, and forage crops, while millet, lentils, barley, and plantain, crops commonly grown in the developing world, received limited CBC research attention. They concluded that much more research was needed for most cropping systems. Furthermore, using a large database of 132 studies at 6759 sites worldwide, Karp et al. [60] modelled natural enemy and pest abundances, predation rates, and crop damage as a function of landscape composition. They found that each of the above indicators exhibited different responses across studies—natural enemy responses were heterogeneous and there was no consistent trend; thus, they concluded that the surrounding non-crop habitat does not consistently improve pest management. And recently, Wyckhuys et al. [61] looked at the role of diversification with 25 legume species, commonly used as intercrops, for natural biological control in agroecosystems. As legumes secrete copious amounts of energy-rich (floral, extra-floral) nectar and provide alternative hosts or prey items for resident natural enemies, their deployment is expected to benefit biological control. However, the mechanistic basis is poorly understood and scientific underpinnings are weak. Their study showed that although natural enemies regularly forage on legumes, the data on interaction linkages are profoundly incomplete due to lack of research.

### 7.1. Role of Field Margins in Pest Management in Agroecosystems

Deliberately managed natural or sown field margins and hedgerows have become a common component of temperate farming systems for maintaining diversity for pollinators and invertebrate food for birds [56]. These plant communities are also expected to support populations of natural enemies of pests of the associated crop. However, their success depends on seasonal synchrony of natural enemies and host prey. Studies of the effectiveness of field margins to enhance cereal aphid control in the UK have been variable [62]. If field margins are to be used effectively to manage pests, more research is needed on their ecology [63]. Segre et al. [64] observed that although field margins increased potential biological pest control in Israel, there was no spillover to fields and farmers lost revenue in most crop types. In a meta-analysis of 235 publications, Mikenda et al. [65] found that field margins enhanced ecosystems services such as natural pest regulation, pollination, and nutrient cycling; however, there were also a number of cases of ineffective pest regulation by field margin vegetation (see Table 1 in ref. [65]), notably aphid vectors of crop viruses. A recent meta-analysis by Crowder et al. [66] found that although field margins were linked to higher abundance and diversity of natural enemies and lower numbers of herbivorous invertebrate pests, very few studies looked at crop damage and yield effects. Clearly, more research is needed to improve the understanding of how effectively field margins can reduce crop losses due to pests. Monitoring numbers of natural enemies and pests in field margins is not sufficient to judge whether they are a viable pest management strategy.

### 7.2. Natural Pest Management in Irrigated Monoculture Rice Systems

Rice is the world’s most important staple food crop, feeding over 3.5 billion people daily, including the majority of the world’s poor (https://www.cgiar.org/research/center/irri/, accessed on 29 March 2024). Irrigated monoculture rice systems are one of the least diverse agroecosystems but have survived and remained stable and productive for over 8000 years in spite of hosting serious rice pests. This stability appears to be based on inherent characteristics of the system [67]. Most importantly, research over four decades has contributed to an in-depth understanding of natural pest management in rice systems.

Irrigated monoculture rice hosts a considerable diversity of generalist predator spiders. There are over 300 species of rice land spiders discovered and described in South and Southeast Asia [68]. Recent studies in Japan have further contributed to the knowledge base of predator spiders [69]. Early in the rice crop, a generalist predator population of spiders feeds on detritus and plankton-feeding insects. This gives predators a head start when rice pests later arrive, including serious pests such as leafhoppers [70]. The decoupling of predator populations from strict dependence on rice pests lends stability to irrigated rice systems [67,71]. More recent research has found that herbivore-induced rice volatiles attract rice field spiders, potentially improving control of rice pests [72]. Barrion [73] summarized key system factors supporting populations of predatory spiders in rice agroecosystems. These included maintaining piles of rice straw around rice fields; avoiding over-grazing, burning, and use of herbicides on rice bunds; and no/judicious use of insecticides against rice pests.

Settele and Settle [74] noted that irrigated rice in subtropical and tropical regions was under-represented in Karp et al.’s [60] study even though it failed to show consistent support for the premise that a non-crop habitat in farming landscapes universally increases biocontrol of crop pests. Notwithstanding, research on natural pest management in rice has shown that surrounding vegetation effects are not major factors determining population patterns of rice pests and their natural enemies [67,71]. More recent studies [75] have found that asynchronous cropping, creating a mosaic of cultivated and temporarily (mostly for short periods) unused fields, provides a continuous supply of resources for predators and parasitoids over space and time, helping natural enemies to avoid spatial and temporal bottlenecks [76]. And, in support of Barrion [73], Horgan [77] pointed out that the abundance of parasitoids and species richness of both parasitoids and predators increased with the structural connectivity of rice bunds. Management of irrigated rice insect pests through the conservation of generalist predator spiders is one of the best researched and understood and widely successful examples of natural pest management in agroecosystems [78].

More research is needed to understand the relationship between crop-associated biodiversity and pest management.

Over 40 years ago, Way [79] stressed that diversity per se cannot be assumed to reduce insect damage in agroecosystems. A decade later, Andow [80], in this context, observed that “*While some of the major twists in the Gordian knot of vegetational diversity can be perceived, we are a long way from unravelling its complexity*”. Furthermore, Wood and Lenné [81] concluded that crop-associated biodiversity can be a mixed blessing for farmers for pest management.

Tscharntke et al. [82] identified possible reasons why crop-associated biodiversity around fields fails to enhance biological pest control. These included the following: pest populations have no effective natural enemies in the region; natural habitat such as woodland is a greater source of pests than natural enemies; crops provide more resources for natural enemies than does natural habitat (as in rice systems); natural habitat is insufficient in amount, proximity, composition, or configuration to provide large enough enemy populations for pest control; and agricultural practices counteract enemy establishment and biocontrol provided by natural habitat due to over-use of insecticides.

It is clear that although the role of crop-associated biodiversity in managing pests is better understood for some crops such as irrigated rice, much research is still needed on other major cropping systems. Currently, the premise that diverse agroecosystems are an important source of predators and parasitoids of crop pests, and as a result will naturally control pests, is not underpinned by sufficient sound knowledge for wide-scale recommendation to farmers for most cropping systems.

## 8. Conclusions

This perspective paper has questioned the increasing promotion of crop diversity as a key method to manage pests in agroecosystems as an alternative to monocultures which produce most of our food. We have shown that environmental or abiotic stresses such as flooding and fire are particularly important in the evolution of monodominance in grasslands, including the wild relatives of important cereals such as rice, wheat, and barley. We have also pointed out that pest pressure on monodominant stands is likely to have increased adaptation to resist pests through the evolution of resistant traits which are now the focus of modern crop breeding. Furthermore, the trans-continental introduction of crops to escape from their indigenous pests, especially into Africa, supported by plant quarantine systems, has facilitated both pest management and increased food production. Finally, using irrigated rice systems as a yardstick for in-depth understanding of crop pest management in the world’s most important food crop, we conclude that considerably more research is needed before crop diversity can be widely recommended for pest management and food production in agroecosystems.

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
