# Peer review of "Crop Diversity in Agroecosystems for Pest Management and Food Production"

_plants, 2024, doi:10.3390/plants13081164_

Round 1

Reviewer 1 Report

Comments and Suggestions for Authors

Review of "Crop diversity in agroecosystems for pest management and food production"

While I am a researcher who looks at functions of biodiversity in a sizeable portion of his work, I welcome an article that challenges some of the notions that are taken for granted in today's literature. It will be useful for readers to ask themselves what we know and what we don't know, and nuance the framing of their study accordingly. Also the references to Daniel Janzen's work are welcome. Young people coming to the field may not be familiar with this name. This being said on the positive side,  I think the current contribution needs improvements, mainly in the definition of terms, the provision of evidence, and the tone. I would recommend avoiding statements that sound like reckonings with David Tilman. This feels inappropriate, and I think it detracts from the core message, which is in my opinion that benefits of monostands, and their natural occurrence, should not be underrated and the benefits of diversity not overrated as is indeed so often done. Detailed comments are below.

Detailed comments

The article is rich in terminology, and I feel terms need definitions to avoid confusion of concepts. I would recommend making a box with definitions. Candidate terms (with line number in brackets) for definition are:

Diverse agroecosystems (9)

Crop diversity (11)

Associated diversity (11)

Monodominance (12) (Monodominance is defined in the text, but rather late, after the term has already been used.)

Monoculture (15)

Trans-continental crop introduction (17)

Diverse agricultural landscape (28)

You may think of more terms to add ...

13, 14 The terms fight and flight response are introduced but hardly used. These terms are related to behaviour of animals in contests; I feel they are not helpful in the context of this paper.

9-11 In relation to the terms "diverse agroecosystems" and "crop diversity" I would note that rice as a crop in a plot and as a crop in its environment is usually a diverse system (due to the water layer and its biota, often small field sizes, and the bunds as a semi-natural habitat), but it's still a monostand of the crop species. That is to say: a field with a crop monostand can be a diverse agroecosystem in itself or it may be part of a diverse agroecosystem (the plot + the bunds or the entire landscape). The way in which the terms are used in these sentences suggests that "diverse agroecosystems" and "crop diversity" are more or less synonymous but for me they are not. The meaning of the authors needs to be made clearer.

25 This -> The

32 "considered to be" is redundant. Omit.

45 "their preferred solution" I think the authors should not underestimate that researchers may not so much "prefer" diversity, but have overlooked the potential of a monostand to be a diverse agroecosystem in itself. I think the authors would agree that the intrinsic biodiversity of a rice crop due to the biota in the water layer is a key reason for the pest suppressiveness of the rice crop. So rice is not a perfect counterexample against benefits of diversity.

47, 50 fight and flight response are unnecessary terms. Suggest to not use the terms. Suggest to focus on the actual mechanisms of, e.g., anti-herbivory chemistry (e.g. tannins) and the intercontinental crop introduction (a kind of predator release – with the herbivore as a plant "predator" that is left behind in the endemic range of the host)

57-65 I feel what is written in these lines is not beliefs but facts. Agricultural systems are in fact made "simple" (perhaps simplified is a misnomer; we do not always know whether what was before agriculture was more complex), they are disturbed (yearly vegetation removal, soil tillage, regular pesticides), and they are nutrient rich (fertilizer). Indeed, agriculture is done with monocultures on fertilized soils. And leaching losses are paramount, though I would agree that "instability" is nonsense (except perhaps in the case of severe soil erosion). It is also true that many crops grown as monostands have high incidence of weeds, pathogens and pests, but I have seen that also in organically grown mixtures, so it is not a unique feature of monostands. The literature is, however, quite clear that ON AVERAGE, mixtures have less weeds, less disease and less pests than monostands. There is quite sufficient literature documenting this, and more will be coming. For a recent (not comprehensive) overview with several useful references, see, e.g., van der Werf & Bianchi (2022) in Outlook on Agriculture. I point to the paper for the references.

74 "It is easy" is easy to say, but a quantitative statement on the prevalence of monostands would be be more informative. I suppose it is not known, but that might be useful to point out. The authors rightly indicate later in the paper that more research on the coping mechanisms of natural monostands would be useful. How do monostands prevent being wiped out by pathogens and herbivores?

83 The authors state that abiotic stress favours monodominance (still not defined). It would be good to note that agriculture aims to take away the limitation of nutrients and water. If that is done, the abiotic stress would be mitigated. Would monodominance then still be favoured?

98 LAM paper? What LAM?

102 The sentence starting with "however" seems out of context. It could follow the text on Daniel Janzen's work on secondary plant chemicals, but it does not fit well here.

101-116 The paragraph needs a concluding statement/conclusion. I accept that cereal ancestors are often monodominants. So?

118-121 I suppose you just want to indicate that these two stresses do the job of favouring monodominance. But are there no other stresses than fire and flooding that can do that? And as far as I know, the American native prairie is fairly species rich. Does it qualify as a monodominated stand or does it not? Is Tilman perhaps influenced by his experience in the Americas? Is he right that agriculture would be simplified if it replaced American prairie? What people think and belief may be strongly influenced by their lived experience.

139 Some of the world's most extensive (and by implication supposedly most stable) ecosystems contain few species. But when researchers advocate diversity, they probably mean increasing the number of species per field from one to a few. So is there a contradiction?

151 Here is the definition of monodominance. It should come earlier.

155-158 I am not sure what you are trying to say here. Do you mean that species that are 100% C, S, or R tend to be mono-dominants? Maybe that is not so strange, because if there is such a strong focus of a species on a specific coping mechanism, then it might have the winning strategy in an environment that is very good and stable (C), very resource limited or toxic (S) or unpredictable and temporary (R).

166-169 Please rethink the sentence. These are indeed characteristics of the green revolution, no?

176 You seem to suggest that Tilman was not paying attention in the workshop. That may not be the case and it may backfire. He may just have made a different synthesis of the workshop than the authors feel is appropriate. I'd suggest to focus on the facts, and remove the references to Tilman as a person.

183-187 I am not convinced the arguments are valid. Persistence of monostands does not mean that they have agriculturally desirable properties. Mere survival is not enough for agricultural crop species. They should produce a yield.

195 There is sufficient evidence that species mixtures have lower disease incidence than monostands. See, e.g. Boudreau in Annu. Rev. Phytopathol. 2013. 51:499–519.

200 Suggest to put "apparently" before "better"

209 I would suggest that the onus of figuring out how monostands cope with herbivores is on those who want to promote monostands, not on those advocating diversified crops or landscapes. And I do not think the current paper details how monostands cope with herbivores. I feel you are merely arguing that they exist and are not unnatural, which is a point clearly made.

214-215 Are these plants with insecticidal secondary metabolites often growing in monostands? Only in that case will the production of insecticidal compounds by these species be related to monodominants. Needs clarification.

217 "cause most of the damage to leaves" is ambiguous. Do you mean they mainly feed on leaves or do you mean that most of the damage to leaves is caused by insects with a narrow host range?

228 I suppose that by "locationally unique" you mean "in the endemic range of a plant species". That is not obvious. And "over short distances" suggests you mean processes at the scale of meters to tens of meters. Please clarify.

241 Deserves strong emphasis in or for what?

250, 251 Inappropriate phrasing. Please correct.

259 Instead of "equals", consider "would confer".

268 Please leave Tilman out of it and focus on the issue.

281 Omit "general"

297 pastures -> pasture grasses?

359 contain -> support? (You want the enemies to leave the habitats that support them and do something useful in the crops)

383-385 I'd omit this sentence or put it elsewhere. I thought that you meant that Settle et al. (1996) has contributed this in-depth understanding, but I think the sentence really refers to what comes in the following paragraphs. As I remember Settle's paper has good ideas, but not much supporting evidence (but perhaps I forgot ...).

387 It is not so clear which generalist population in the previous sentences you are referring to. Avoid the word "this".

414, 415 I doubt whether this presumption is correct. Another possibility is that researchers from temperate regions are not sufficiently aware of the situation in the rice agroecosystem to confidently refer to it. It may reflect a geographic bias, with Asian researchers more likely referring to studies in rice than American or European researchers. Just another possibility. It does not necessarily represent an anti-monostand bias.

Comments on the Quality of English Language

None

Author Response

Reviewer 1: "Crop diversity in agroecosystems for pest management and food production"

While I am a researcher who looks at functions of biodiversity in a sizeable portion of his work, I welcome an article that challenges some of the notions that are taken for granted in today's literature. It will be useful for readers to ask themselves what we know and what we don't know, and nuance the framing of their study accordingly. Also the references to Daniel Janzen's work are welcome. Young people coming to the field may not be familiar with this name. This being said on the positive side, I think the current contribution needs improvements, mainly in the definition of terms, the provision of evidence, and the tone. I would recommend avoiding statements that sound like reckonings with David Tilman. This feels inappropriate, and I think it detracts from the core message, which is in my opinion that benefits of monostands, and their natural occurrence, should not be underrated and the benefits of diversity not overrated as is indeed so often done. Detailed comments are below.

Thank you for the refreshingly positive comments on the paper as well as the recommendation for improvements. We hope we have constructively addressed the suggested revisions with improvements. Most statements referring to Tilman have been removed.

Detailed comments

The article is rich in terminology, and I feel terms need definitions to avoid confusion of concepts. I would recommend making a box with definitions. Candidate terms (with line number in brackets) for definition are:

Diverse agroecosystems (9)

Crop diversity (11)

Associated diversity (11)

Monodominance (12) (Monodominance is defined in the text, but rather late, after the term has already been used.)

Monoculture (15)

Trans-continental crop introduction (17)

Diverse agricultural landscape (28)

You may think of more terms to add ...

This is a good idea. We have inserted a Box with definitions of commonly used terms. We changed all references to “diverse agricultural landscapes” to “diverse agroecosystems” to avoid confusion. We understand the meaning of ‘monostand’ but it is not used much in the non-ecological literature. Monoculture is more widely used so we have remained with this term.

13, 14 The terms fight and flight response are introduced but hardly used. These terms are related to behaviour of animals in contests; I feel they are not helpful in the context of this paper.

We have redefined the fight and flight responses to: genetic resistance response and escape response throughout the paper.

9-11 In relation to the terms "diverse agroecosystems" and "crop diversity" I would note that rice as a crop in a plot and as a crop in its environment is usually a diverse system (due to the water layer and its biota, often small field sizes, and the bunds as a semi-natural habitat), but it's still a monostand of the crop species. That is to say: a field with a crop monostand can be a diverse agroecosystem in itself or it may be part of a diverse agroecosystem (the plot + the bunds or the entire landscape). The way in which the terms are used in these sentences suggests that "diverse agroecosystems" and "crop diversity" are more or less synonymous but for me they are not. The meaning of the authors needs to be made clearer.

We hope that we have now made the terms clearer with the definitions. The section on irrigated monoculture rice discusses the non-crop diversity e.g. spiders in such systems.

25 This -> The Revised

32 "considered to be" is redundant. Omit. Revised

45 "their preferred solution" I think the authors should not underestimate that researchers may not so much "prefer" diversity, but have overlooked the potential of a monostand to be a diverse agroecosystem in itself. I think the authors would agree that the intrinsic biodiversity of a rice crop due to the biota in the water layer is a key reason for the pest suppressiveness of the rice crop. So rice is not a perfect counterexample against benefits of diversity.

The “preferred solution” has now been removed. We have also revised the bullets related to other revisions in the paper. We think the reviewer is right in pointing out that researchers have overlooked the potential of a monoculture. We would like to use the term “monoculture” throughout as it is more widely known outside of ecology.

47, 50 fight and flight response are unnecessary terms. Suggest to not use the terms. Suggest to focus on the actual mechanisms of, e.g., anti-herbivory chemistry (e.g. tannins) and the intercontinental crop introduction (a kind of predator release – with the herbivore as a plant "predator" that is left behind in the endemic range of the host)

We have changed the terns throughout to “genetic resistance response” and the “escape response”

57-65 I feel what is written in these lines is not beliefs but facts. Agricultural systems are in fact made "simple" (perhaps simplified is a misnomer; we do not always know whether what was before agriculture was more complex), they are disturbed (yearly vegetation removal, soil tillage, regular pesticides), and they are nutrient rich (fertilizer). Indeed, agriculture is done with monocultures on fertilized soils. And leaching losses are paramount, though I would agree that "instability" is nonsense (except perhaps in the case of severe soil erosion). It is also true that many crops grown as monostands have high incidence of weeds, pathogens and pests, but I have seen that also in organically grown mixtures, so it is not a unique feature of monostands. The literature is, however, quite clear that ON AVERAGE, mixtures have less weeds, less disease and less pests than monostands. There is quite sufficient literature documenting this, and more will be coming. For a recent (not comprehensive) overview with several useful references, see, e.g., van der Werf & Bianchi (2022) in Outlook on Agriculture. I point to the paper for the references.

We have modified the terminology re beliefs. Most Tilman criticisms have now been omitted and the section is written in a more straight forward way including adding the suggested reference – thank you. I should have added it anyway as I have previously read the paper.

74 "It is easy" is easy to say, but a quantitative statement on the prevalence of monostands would be more informative. I suppose it is not known, but that might be useful to point out. The authors rightly indicate later in the paper that more research on the coping mechanisms of natural monostands would be useful. How do monostands prevent being wiped out by pathogens and herbivores?

We did search for quantitative data on the prevalence of natural monostands/monocultures without success. There is some data on forest trees but not much else. We have revised with link to following section which provides many examples of wild relatives of mainly cereals.

83 The authors state that abiotic stress favours monodominance (still not defined). It would be good to note that agriculture aims to take away the limitation of nutrients and water. If that is done, the abiotic stress would be mitigated. Would monodominance then still be favoured?

Monodominance has now been defined in Box. Interesting question – are there examples where wild relatives of crops evolved in monocultures without nutrient or water stress? Our context is more in the evolution of monodominance in the wild relatives of the founder cereals under flood or fire depending on the cereal. Unfortunately there is very little information for other crop groups. In the case of rice, as few plants are adapted to growing under flooded conditions, it is unlikely that the wild relatives of rice would have faced much competition.

98 LAM paper? What LAM?

A paragraph has now been added to explain LAM – large-seed, awns and monodominance

102 The sentence starting with "however" seems out of context. It could follow the text on Daniel Janzen's work on secondary plant chemicals, but it does not fit well here.

Thank you – sentence has now been removed.

101-116 The paragraph needs a concluding statement/conclusion. I accept that cereal ancestors are often monodominants. So?

Sentence added.

118-121 I suppose you just want to indicate that these two stresses do the job of favouring monodominance. But are there no other stresses than fire and flooding that can do that? And as far as I know, the American native prairie is fairly species rich. Does it qualify as a monodominated stand or does it not? Is Tilman perhaps influenced by his experience in the Americas? Is he right that agriculture would be simplified if it replaced American prairie? What people think and belief may be strongly influenced by their lived experience.

Addressed as far as possible.

139 Some of the world's most extensive (and by implication supposedly most stable) ecosystems contain few species. But when researchers advocate diversity, they probably mean increasing the number of species per field from one to a few. So is there a contradiction?

Yes, there are many contradictions in the diversity – stability debate. We have changed this paragraph to some extent based on Reviewer 2’s comments on the Dover and Talbot reference. It is true that there are examples of ecosystems with few species that are stable - some under significant abiotic stresses e.g. bogs, heathland, mangroves etc. and there are other very species diverse ecosystems such as coral reefs and tropical forests which are highly diverse but very fragile. Many researchers advocating diversity do not define how much diversity they are advocating.

151 Here is the definition of monodominance. It should come earlier.

We have now provided this in the Box. It is useful to reinforce what we mean by monodominance.

155-158 I am not sure what you are trying to say here. Do you mean that species that are 100% C, S, or R tend to be mono-dominants? Maybe that is not so strange, because if there is such a strong focus of a species on a specific coping mechanism, then it might have the winning strategy in an environment that is very good and stable (C), very resource limited or toxic (S) or unpredictable and temporary (R).

Slightly revised.

166-169 Please rethink the sentence. These are indeed characteristics of the green revolution, no?

We have revised this.

176 You seem to suggest that Tilman was not paying attention in the workshop. That may not be the case and it may backfire. He may just have made a different synthesis of the workshop than the authors feel is appropriate. I'd suggest to focus on the facts, and remove the references to Tilman as a person.

We have revised this.

183-187 I am not convinced the arguments are valid. Persistence of monostands does not mean that they have agriculturally desirable properties. Mere survival is not enough for agricultural crop species. They should produce a yield.

The section from L160-187 has been rewritten and we hope it addresses the three issues raised.  

195 There is sufficient evidence that species mixtures have lower disease incidence than monostands. See, e.g. Boudreau in Annu. Rev. Phytopathol. 2013. 51:499–519.

This paper is about intercrops and we have made it clear in the definitions section that the paper is focused on monocultures. Some issues in regard to intercrops and disease were addressed in Lenné (2023). There are three meta-reviews of crop varietal mixtures and diseases – mostly covering cereal mixtures – which have shown less disease in mixtures but unfortunately very few of these studies also measured yield loss. These mixtures are monocultures – at least by the original definition of the term – a monospecific stand. Apart from reviews of diseases in intercrops, I am not aware of literature on diseases in polycultures.

200 Suggest to put "apparently" before "better"

Revised

209 I would suggest that the onus of figuring out how monostands cope with herbivores is on those who want to promote monostands, not on those advocating diversified crops or landscapes. And I do not think the current paper details how monostands cope with herbivores. I feel you are merely arguing that they exist and are not unnatural, which is a point clearly made.

Revised

214-215 Are these plants with insecticidal secondary metabolites often growing in monostands? Only in that case will the production of insecticidal compounds by these species be related to monodominants. Needs clarification.

Revised and hopefully clarified

217 "cause most of the damage to leaves" is ambiguous. Do you mean they mainly feed on leaves or do you mean that most of the damage to leaves is caused by insects with a narrow host range?

Revised

228 I suppose that by "locationally unique" you mean "in the endemic range of a plant species". That is not obvious. And "over short distances" suggests you mean processes at the scale of meters to tens of meters. Please clarify.

Hopefully clarified

241 Deserves strong emphasis in or for what?

250, 251 Inappropriate phrasing. Please correct.

259 Instead of "equals", consider "would confer".

268 Please leave Tilman out of it and focus on the issue.

281 Omit "general"

The section related to these concerns has been rewritten and restructured. We have addressed all of these issues including Tilman.

297 pastures -> pasture grasses?

Revised

359 contain -> support? (You want the enemies to leave the habitats that support them and do something useful in the crops)

Revised

383-385 I'd omit this sentence or put it elsewhere. I thought that you meant that Settle et al. (1996) has contributed this in-depth understanding, but I think the sentence really refers to what comes in the following paragraphs. As I remember Settle's paper has good ideas, but not much supporting evidence (but perhaps I forgot ...).

Revised

387 It is not so clear which generalist population in the previous sentences you are referring

  1. Avoid the word "this".

Revised

414, 415 I doubt whether this presumption is correct. Another possibility is that researchers from temperate regions are not sufficiently aware of the situation in the rice agroecosystem to confidently refer to it. It may reflect a geographic bias, with Asian researchers more likely referring to studies in rice than American or European researchers. Just another possibility. It does not necessarily represent an anti-monostand bias.

Perhaps not by all but by some of the most vocal in influential debates – we have deleted the sentence.

Reviewer 2 Report

Comments and Suggestions for Authors

In this manuscript the authors put forward the concept of vegetation monodominance to challenge the diversity-stability hypothesis with respect to pest management in alternative and conventional agricultural systems. In their analysis, the authors fail to properly characterize monodominant systems in natural systems, portraying them as being dominated by a single species, and thus equating them to the planting of monocultures in modern agriculture. No information, based on the literature, is provided with respect to the actual plant species and germplasm diversity in these systems, as compared to the crop species and narrow genetically-uniformity observed in modern monoculture systems. In addition, the authors need to clarify under what ecological zones are monodominant species most prevalent in nature, and whether these zones match or are the same as the ecological zones of the major global agricultural production areas (such as in the USA, Latin America, or in Asia).

Overall, the authors propose an interesting hypothesis, but it is based on selected quotes (provided without context); and following a logic of false-equivalence (natural vs agricultural systems), without providing the proper justification or characterization-- for justifying this system comparison. The perspective should also refer to more recent citations, as it relies on some of the more earlier ecological literature.

As part of their analysis, the authors employ the case of cereals as a model to support their hypothesis. Even if their hypothesis holds, the authors fail to describe whether the monodominant hypothesis applies to all crop species in general, or only for cereals. The authors thus need to justify the use of cereals as model crops for their analysis, and indicate whether their observations can be generalized to all crop species and agricultural settings.

As part of their analysis, to support their hypothesis, the authors cite instances in which diversity-stability experiments have not shown reduced pest incidence or activity of natural enemies. However, it is well established in the literature that arthropod species respond differentially to habitat modification strategies, depending on variables such as ability to disperse and host-range.

As part of their analysis, the authors have failed to refer to the extensive literature on intercropping or polycultures, and their impact on pest dynamics, when compared to their component monocultures-- to determine whether the literature on pest dynamics in polycultures supports their hypothesis on monodominance.

As part of their argument, to support the monodominance hypothesis, the authors indicate that crops do better in introduced regions. However, the authors fail to mention the frequent development of pest outbreaks and epidemics in these introduced regions (e.g. Puccinia outbreaks in wheat; Bipolaris or southern corn leaf blight; phytophthora in potato), and how do these epidemics that occur in introduced regions reconcile with their case on monodominance. Thus, extensive pesticide applications are still required for the production of crops in ‘introduced’ regions, and thus, the authors need to reconcile this practice, with their hypothesis.

Additional comments on the text include,

L 60-63, This sentence does not stand on its own, and is thus unclear. Does this sentence represent a statement made by Tilman (1999), or by the authors?

L 63, Re: “all of which did occur.” This phrase is unclear, When did this occur? It would be more precise to cite references confirming that the purported impacts of monoculture agriculture, as indicated by Tilman, have indeed been reported in the scientific literature.

L 67-69, An analysis on the vegetational simplification of modern agricultural systems cannot be considered a ‘concept’ of agroecology, but rather an analysis of modern ag systems, which then calls for the design of alternative or ecologically-based production systems.

L 73, The authors are making a strong statement, and to back it up it is important to clarify if Altieri has actually made a claim to the “the universality of `nature's diversity’.” It is important that the authors make a clear distinction of ecological cases where the monodominant vegetation of wild relatives has been reported, with proper first-source citations. The authors should also clarify whether the extent to which the monodominant wild relative systems that they cite, have the same level of biological diversity (above and below ground), as compared to the level of genetic/biological uniformity observed in expansive conventional monocultures that receive ongoing high rates of chemical and fertilizer applications, and that experience intensive tillage, soil disturbance, erosion, and compaction. 

L 74-75, Re: “It is easy to find monodominant vegetation under a range of ecological conditions.” Again, the authors should provide first source citations that document and characterize the ecology of ‘monodominant vegetation under a range of ecological conditions’, and to cite comparative first-source studies that confirm that the observed germplasm diversity of the wild relatives and field biodiversity (above and below ground) in these ‘monodominant’ systems is similar to the ones observed under large-scale intensive monoculture systems. 

L 75-76, The phrase under parentheses is unclear. 

L 77-78, This sentence is unclear. Is this a quote from the source, or a statement by the authors? Need to clarify.

L 78-89, Here the authors criticize the statement “It was not nature's way to allow large expanses of land to be planted to a single crop” and in turn make their case for Monodominance to ‘prove’ that the statement is wrong, which is thus self-contradictory. The authors need to report the extent of Monodominance, as a percent (by area) of all natural terrestrial ecosystems, and as a percent (by area) of all traditional/indigenous agricultural systems. What percent of all cereal species (small and large grains), and what percent of ALL edible wild relative plant species, such as tropical root crops, have shown the syndrome or trait of Monodominance under natural ecological settings?

L 89, The reference cited only refers to cereal crops. Does the statement also refer to non-cereal crops? The scientific literature is extensive with studies showing that intercropping with multiple species is often a successful strategy to minimize the risk of entire crop losses from biotic or abiotic stress.

L 89-90, Why were these two types of environmental stresses selected? Are these types of stress representative or relevant in agricultural settings?

L 115-116, Again, it would be helpful to cite ecological studies comparing the vegetational and biological diversity in these monodominant systems observed in nature as compared to modern intensive agricultural monocultures.

L 122-124, Awkward, please rephrase.

L 124, What is “species-diverse-species vegetation”? Please clarify.

L 125-130, It is well understood that different pest species react differently to varied different vegetational diversification settings and that behavioral or dispersal models cannot be applied universally. Under different vegetational settings, such as under forestry, diversity may also have a differential impact on stability. However, the observed variability in response to diversity, and a lack of ‘universality’ don’t in themselves invalidate the theory that diversified systems may increase stability under some or most agroecological settings.

L 146, Re: “In fact, these communities are among the most fragile.” No citations are provided to back up such a strong statement. For one example on the fragility of large-scale tree monocultures, based on a large-scale multi-year experiment, please refer to the experience of the Great Green wall of China (Three-North Shelter Forest Program). 

L 146-148, How can researchers “conclude an idea”? And how does an “idea” and a “notion” prove a scientific hypothesis? The authors seem to be conflating experimental results with the personal opinion of some scientists, based on “notions,” “ideas” and on a selective and narrow analysis of the scientific literature.

L 149-150, It would be helpful, for the reader, to provide an ecological characterization of a monodominant system. Other than the >60% of the monodominant plant species (percent ground cover?), how many other species are found in these systems, and how would this compare to the plant vegetational and germplasm uniformity observed under industrial monoculture systems? For instance, in natural grassland prairies in the USA over 100 plant species may be found in small areas (<2 Ha), and over 250 species in large areas (of 450 Hectares) (Risser, P.G., 1988. Diversity in and among grasslands (pp. 176-180). In: Biodiversity. Washington, DC: National Academy of Sciences).

L 169-170, The authors need to substantiate the statement that traditional paddy rice cultivation conducted over hundreds of years, following natural methods of production, is equivalent to green revolution production practices including the reliance on genetically uniform cultivars (narrow germplasm base), and the application of high rates of fossil fuel-based chemical pesticides and fertilizers. The authors also need to substantiate the idea that the genetic background or genetic pool of modern production systems (based on planting one cultivar over extended areas) is equivalent to the genetic background of traditional or indigenous production systems. While modern production systems rely on a very narrow genetic pool, traditional farming communities, within the same geographical area (e.g. Asia, Africa, Latin America), often relied on a large number of varieties and landraces, representing a wider genetic pool.

L 186-187, The authors are making a false-equivalence comparison, by comparing two separate systems that are unrelated to each other. By stating that Monodominance systems occur in some instances in nature (systems that the authors have failed to properly characterize in terms of their vegetational diversity, and percent area coverage in relation to all terrestrial ecosystems), the authors claim that industrial monoculture systems are also ‘stable’ which are contradicted by an extensive scientific record that has documented the instability of agricultural monocultures (such as their susceptibility to pest outbreaks, climatic events and broken nutrient cycles).

L 220-223, Here, to support their claim, the authors should cite specific examples in which breeders have developed pest resistant varieties with anti-feeding compounds. The fact that crop breeders aim to develop resistant varieties does not support the authors’ claim, that resistance develops naturally under monocultures, especially with varieties that have a uniform genetic background— as is common in commercial crop monocultures.

L 224, In this section the authors need to explain why extensive tillage and pesticide applications are still required for the production of introduced cereal crops (as compared to the center of origin), even though they have reached ‘Monodominance’ by not having to confront the presence of “coevolved pests and diseases”. The authors also need to explain the frequent occurrence of crop epidemics, such as the potato leaf blight epidemic in Ireland during the 1800s, even though these epidemics have occurred in “Introduced areas.”

L 256-257, This sentence is unclear, poor syntax. Also, provide citations, such as for cereals, root crops, and indigenous leafy crops-- to back up the statement.

L 258-271, The authors here, again, have to reconcile the fact that a considerable amount of external inputs, adapted tillage, and pesticides are required to grow soybeans in Latin America even though pests and diseases have been “left behind an oceanic barrier.” To back-up their claims about Monodominance, the authors should provide comparative data, with citations, on the number of major pests and the levels of soybean crop losses both from China (center of origin) and Latin America (introduced species) when grown under monoculture and WITHOUT pesticide applications.

274-276, Poor sentence structure.

272-280, With their claim that indigenous crops are “disadapted,” the authors need to reconcile the apparent contradictions, for example, with the successful production of indigenous crops such as rice in China, potatoes in the Andes, corn in the Americas, and root crops in Asia and the Pacific-- over hundreds of years, without the need of external chemical inputs— in contrast to the chemical intensive systems observed today for the production of introduced species in countries such as the USA (or soybeans in Latin America).

L 224-319, The purpose of section 5 is unclear. Yes, initially, after introduction, crops may be grown without many of the major pests found in their center of origin. However, as indicated in L 302, eventually sporadic pest outbreaks become the norm with introduced crops, due in part to their vulnerability when grown under monoculture systems. Major pest epidemics have been experienced with introduced crops in the USA, such as with corn, wheat, and many vegetable species. Monodominance is then maintained through the reliance on high rates of external chemical inputs.

L 320-354, In this section the authors cite only one study published over the past 5 years. The authors support their case with respect to the effect of vegetational diversification, based on undefined and non-specific generalizations, such as using the terms “inconsistent” or claiming that “it is not necessarily true.” It would be more valuable to report the total number of published studies that have shown lower pest levels due to diversification vs. the number of studies that have shown no effect on diversification. It is well known that some pests species are better managed under diversified systems, than other species (due to variables such as their host range and dispersal behavior)— and thus it would be misleading to make blanket assessments without characterizing the type of pests, under discussion.

L 384, It would be helpful for the authors to characterize what they mean by rice “monoculture” based on the region of production. For example, in Asia, the authors should describe the size of the plots in small vs large farms, the incidence and type of vegetation in the field borders, the incidence of rotations with other crops, as well as the vegetational diversification from a landscape perspective (such as the ‘mosaic’ cropping patterns as described in L 405)— and compare these systems with the larger scale monocultures found under industrialized systems, such as in the USA.

L 401-404, This statement uses a very strong statement from a scientific perspective (“conclusively”), by citing two studies published over 25 years ago. To back up their statement, the authors again use ambiguous language, i.e. “not major factors,” which leaves the reader with the question of what is exactly meant by “not major factors” (only 10, 20 or 48% contribution to natural enemy activity?).

L 414-415, Again, it is ambiguous or unclear to define the rice systems in Asia as monocultures, if the authors have failed to appropriately characterize the systems, including the level of biological diversity within and around the paddies (such as species of algae, azolla, aquatic life, and non-crop vegetation within and around the production fields). Why not provide comparative data with rice monocultures in the USA, as well as its production under dry land or upland conditions, rather than selectively focusing on only one type of production system?

L 436, It is misleading for the authors to claim that, according to ¬¬¬-proponents, the buildup of natural enemy activity via crop diversification should be sufficient to control important pests “without the need for further intervention.” It is well understood by pest management specialists that a multiple number of management tactics are required as part of an integrated pest management program, and that it is ineffective or shortsighted to rely on only a single management practice. Thus, the use of crop diversification to build up the activity of beneficial organisms, or to deter pest infestations, would be considered only one aspect of the overall pest management program, along with improved soil quality, crop sanitation, organic mulches, resistant varieties, among other cultural management strategies.

L 453, References, The authors should follow Plants-MDPI literature citation guidelines.

Additional edit suggestions/questions are included in the attached copy of the manuscript.

/////

Comments on the Quality of English Language

The English is adequate, but requires revisions and clarifications in several instances (please see attached).

Author Response

Reviewer 2

General comments by authors: This paper is a perspective paper and not a review. The main objective of a perspective paper is to present an opinion/perspective supported by 82 selected peer-reviewed publications. The reviewer may not agree with our perspective or our choice of citations to support it however the paper should not be reviewed on the basis of what the reviewer would have written on the topic which clearly disagrees with our perspective.

In this manuscript the authors put forward the concept of vegetation monodominance to challenge the diversity-stability hypothesis with respect to pest management in alternative and conventional agricultural systems. In their analysis, the authors fail to properly characterize monodominant systems in natural systems, portraying them as being dominated by a single species, and thus equating them to the planting of monocultures in modern agriculture. No information, based on the literature, is provided with respect to the actual plant species and germplasm diversity in these systems, as compared to the crop species and narrow genetically-uniformity observed in modern monoculture systems. In addition, the authors need to clarify under what ecological zones are monodominant species most prevalent in nature, and whether these zones match or are the same as the ecological zones of the major global agricultural production areas (such as in the USA, Latin America, or in Asia).

Vegetation monodominance is not a “concept” – it is a reality. I suggest the author consults the literature. The paper is not specifically about challenging the diversity-stability hypothesis although by focussing on the role of monocultures, it does so directly. And we are not the first authors to do so. The original paper describing the hypothesis by Janzen has been cited over 5000 times and there are many citations among these which question its universality. Even Janzen himself questioned it soon after the paper was published because he realized that monocultures/monodominance was common in nature – see Janzen (1974, 1977) cited in this paper.

We have added a Box of definitions which was also suggested by Reviewer 1. It now provides clarity on what we mean by crop diversity, monodominance, monocultures etc. By definition monodominant vegetation is characterized by 60% or more of the same species. We do not mention “monodominant systems” – whatever these are - anywhere in the paper. The paper contains many examples of monodominant grasses, monodominant wild relatives of the founder cereals, monodominant trees etc. We have also added more references to provide even more examples. As monodominant species are found throughout most agroecological zones in both tropical and temperate regions – whether they match major global production areas seems unnecessary especially as we are not focussing on or making specific recommendations in any agroecological zone or country. 

This reviewer does not understand what a monoculture is – a monospecific stand only with no specification of time or definitive space or management practices – these are not part of the original and recognised definition of a monoculture.

Overall, the authors propose an interesting hypothesis, but it is based on selected quotes (provided without context); and following a logic of false-equivalence (natural vs agricultural systems), without providing the proper justification or characterization-- for justifying this system comparison. The perspective should also refer to more recent citations, as it relies on some of the more earlier ecological literature.

Many of the quotes were removed during the preparation of the paper. There are very few in the submitted paper. We are not following false equivalence logic in comparing natural and agricultural systems. We are putting forward a perspective on crop diversity in agroecosystems and showing with many examples that monoculture natural vegetation, which is common in nature, could potentially provide useful strategies for future pest management and food production in crops with more research to understand its function. We provide the best example for the founder cereals – wheat and barley in the fertile crescent and rice in Asia – which from the substantial published literature irrefutably evolved from their monoculture wild ancestors (see Wood and Lenné, 2018 published in the Proceedings of the Royal Society B with over 80 reference citations). We are especially highlighting the older literature as it contains important ecological thinking relevant to this perspective e.g. Janzen which unfortunately has been forgotten by current researchers. Reviewer 1 was pleased that we had revived some of the older literature.

As part of their analysis, the authors employ the case of cereals as a model to support their hypothesis. Even if their hypothesis holds, the authors fail to describe whether the monodominant hypothesis applies to all crop species in general, or only for cereals. The authors thus need to justify the use of cereals as model crops for their analysis, and indicate whether their observations can be generalized to all crop species and agricultural settings.

We have addressed this in the above point and provided the evidence. We have not suggested that the founder cereals are a model for all crops. We have also indicated that it most probably applies to other cereals such as oats, sorghum, millets etc. if large-seeded and awned but there is limited evidence to support its application elsewhere. We have clarified this issue.

As part of their analysis, to support their hypothesis, the authors cite instances in which diversity-stability experiments have not shown reduced pest incidence or activity of natural enemies. However, it is well established in the literature that arthropod species respond differentially to habitat modification strategies, depending on variables such as ability to disperse and host-range.

The paper is not about the diversity-stability hypothesis per se on which there is a huge and complex literature – more than 5000 citations of the original hypothesis - which is beyond the scope of this paper. Also there are a significant number of papers which challenge it including some by Janzen. We agree with the reviewer that arthropods respond differently in different agroecosystems. The main objective of the section on crop-associated diversity in the field is to show that several meta-studies, in some cases analyses of many studies, had provided sufficient examples to question the widespread belief that diversity in cropping systems is sufficient to confer pest management. Several of the meta-analyses cited did show reduced pest incidence and higher activity of natural enemies but a sizable proportion did not. Several references also noted that more research is needed to underpin the science.  

As part of their analysis, the authors have failed to refer to the extensive literature on intercropping or polycultures, and their impact on pest dynamics, when compared to their component monocultures-- to determine whether the literature on pest dynamics in polycultures supports their hypothesis on monodominance.

We have now indicated in the definitions section and Box 1 what we are including and not including. Several papers e.g. Lenné and Wood (2022) and Lenné (2023) addressed varietal mixtures and intercrops. Wyckhuys et al. (2023) – cited in this paper - is specifically on pest management in 25 legume crops in intercrops. We talk about diverse agroecosystems which includes polycultures. As polycultures vary, it was beyond the scope of this paper to look in detail at the many different polycultures and their pest dynamics. Again we are putting forward a perspective on monocultures and not a comparison between monocultures and polycultures. It is not a review.

As part of their argument, to support the monodominance hypothesis, the authors indicate that crops do better in introduced regions. However, the authors fail to mention the frequent development of pest outbreaks and epidemics in these introduced regions (e.g. Puccinia outbreaks in wheat; Bipolaris or southern corn leaf blight; phytophthora in potato), and how do these epidemics that occur in introduced regions reconcile with their case on monodominance. Thus, extensive pesticide applications are still required for the production of crops in ‘introduced’ regions, and thus, the authors need to reconcile this practice, with their hypothesis.

The issue is explored in some detail in Lenné and Wood (2022) which is cited. I have cross-referenced it to the appropriate section of the paper. We would only be repeating the details here. I have extracted some salient points from this paper below. It is not appropriate in a perspective paper to cut and paste sections of already published papers by the same authors. It is interesting that apart from the rust outbreaks, the reviewer mentions the Southern Corn Leaf blight which occurred in the USA 50 years ago and the potato blight epidemic which occurred in Europe and especially Ireland 180 years ago. If pest outbreaks and epidemics are frequent, then when were the other examples?

  1. Pest and disease outbreaks and epidemics are found in introduced crops but not as frequently as in the past due to quarantine regulations and the expanded and successful use of crop resistance. As we have indicated in the paper crop introduction is not fail safe and breakthroughs do occur especially for wind borne pathogens such as Puccinia rusts. But even with the last outbreak of Ug 99, resistance has already been identified in the landrace gene pool and is being bred into existing productive varieties.
  2. Southern corn leaf blight is NOT a disease of an introduced crop as maize is native to the Americas. It was also a very unusual disease outbreak. It resulted from over-reliance on Texas cytoplasmic male sterile lines (cms-T) and an unpredictable mutation in Bipolaris maydis which had previously been a minor disease of maize (Bruns, 2017). Although 15% of the crop was destroyed in one year, use of cms-T was replaced by detasseling the female parent and in the year following the leaf blight, maize secured record yields.
  3. In the 1840s, late blight (Phytophthora infestans) of potato was far more widespread than Ireland. In 1842, late blight was introduced from the Toluca valley in Mexico, where it was common on both wild and cultivated Solanum spp., to the eastern USA. By 1845, blight had spread to Europe through Poland, Germany, Belgium, France and England under favourable weather conditions with every variety of potato being attacked (Large, 1940). There was no resistance in the entire potato gene pool in Europe to this new-encounter pathogen. In Ireland, the reliance of the rural poor on potatoes as a staple food meant that blight affected the Irish more than the rural poor in other countries. Most importantly, the social regime in Ireland (absentee landowners) and the political regime in England (protection of home agriculture from imported food especially cereals) made a significant contribution to the famine in Ireland (Large, 1940). It is noted by Goss et al. (2014) that this is one of the few examples of a pathogen with a known origin that is secondary to the origin of its current major host. As a re-encounter disease, whether the crop was introduced or native, it would have succumbed to the disease

Finally, although pesticides are used, extensive pesticide use is not needed to produce these crops in introduced regions as plant breeding for resistance successfully protects them. 

Additional comments on the text include:

L 60-63, This sentence does not stand on its own, and is thus unclear. Does this sentence represent a statement made by Tilman (1999), or by the authors?  

Removed

L 63, Re: “all of which did occur.” This phrase is unclear, When did this occur? It would be more precise to cite references confirming that the purported impacts of monoculture agriculture, as indicated by Tilman, have indeed been reported in the scientific literature.

Removed

L 67-69, An analysis on the vegetational simplification of modern agricultural systems cannot be considered a ‘concept’ of agroecology, but rather an analysis of modern ag systems, which then calls for the design of alternative or ecologically-based production systems.

Removed

L 73, The authors are making a strong statement, and to back it up it is important to clarify if Altieri has actually made a claim to the “the universality of `nature's diversity’.” It is important that the authors make a clear distinction of ecological cases where the monodominant vegetation of wild relatives has been reported, with proper first-source citations. The authors should also clarify whether the extent to which the monodominant wild relative systems that they cite, have the same level of biological diversity (above and below ground), as compared to the level of genetic/biological uniformity observed in expansive conventional monocultures that receive ongoing high rates of chemical and fertilizer applications, and that experience intensive tillage, soil disturbance, erosion, and compaction. 

Altieri “universality” removed

First source citations for monodominant vegetation have been provided in a number of papers cited in this paper and this perspective paper is not an appropriate place to repeat already published citations of the first source citations. See Wood (2011); Wood and Lenné (2018) and Lenné and Wood (2022)

Discussion on below-ground biodiversity in monocultures, fertilizer use and soil preparation are beyond the scope of this paper – it is about pest management which has been made clear by the title. It is also noted by the authors that the reviewer is using the negative rhetoric about some aspects of the management of monocultures to criticise the paper. We have not referred to such management regimes as they are outside the scope of this paper.

L 74-75, Re: “It is easy to find monodominant vegetation under a range of ecological conditions.”  Again, the authors should provide first source citations that document and characterize the ecology of ‘monodominant vegetation under a range of ecological conditions’, and to cite comparative first-source studies that confirm that the observed germplasm diversity of the wild relatives and field biodiversity (above and below ground) in these ‘monodominant’ systems is similar to the ones observed under large-scale intensive monoculture systems. 

As noted above, first source citations have already been published – this perspective paper should not cut and paste what has already been published but cite the reference where it has been published which we have done. The similarity or not between the ecological settings of monodominant stands of the wild relatives of the founder cereals and modern large scale monoculture systems is outside the scope of this paper. This is like asking us to assess the similarity or not between the ecological settings of humans of more than 10,000 years ago when the wild relatives of the founder cereals were being gathered (that is pre-domestication) with those of today’s humans. It would be meaningless. Also, the reviewer seems to think that farmers globally only grow one type of monoculture – modern industrial monocultures with substantial inputs. Monoculture agriculture is practised by millions of farmers globally on small, medium and large farms and with a very wide range of inputs from none to considerable. It also produces most of the food which feeds more than 8 billion people (in spite of the unspecified frequent pest and disease epidemics). This was addressed in the definitions added to the paper (see Box 1).

L 75-76, The phrase under parentheses is unclear. 

Corrected

L 77-78, This sentence is unclear. Is this a quote from the source, or a statement by the authors? Need to clarify.

Corrected and elaborated.

L 78-89, Here the authors criticize the statement “It was not nature's way to allow large expanses of land to be planted to a single crop” and in turn make their case for Monodominance to ‘prove’ that the statement is wrong, which is thus self-contradictory. The authors need to report the extent of Monodominance, as a percent (by area) of all natural terrestrial ecosystems, and as a percent (by area) of all traditional/indigenous agricultural systems. What percent of all cereal species (small and large grains), and what percent of ALL edible wild relative plant species, such as tropical root crops, have shown the syndrome or trait of Monodominance under natural ecological settings?

This is beyond the scope of the paper. We have already scoured the literature for information on the ecological settings of the evolution of other crops groups and it is not available. Again this is not necessary as we have made it clear now that the information is available for the founder cereals.

L 89, The reference cited only refers to cereal crops. Does the statement also refer to non-cereal crops? The scientific literature is extensive with studies showing that intercropping with multiple species is often a successful strategy to minimize the risk of entire crop losses from biotic or abiotic stress.

We are specifically referring to crop domestication and not cropping systems such as intercrops. We have also added, associated with the definitions Box, that we are not focussing on intercrops in this paper. However we do refer to the Wyckhuys et al (2023) paper which looked at 25 legume crops in intercropping systems and concluded that much more research is needed to show that such crops benefit natural enemies to reduce pest damage. The other benefits of intercrops e.g. nitrogen for the companion crops are not disputed or mentioned in this paper – see Lenné for further discussion on this.

L 89-90, Why were these two types of environmental stresses selected? Are these types of stress representative or relevant in agricultural settings?

It is clear from what we said that fire was important in the ecological settings of the domestication of wheat and barley and flood was important in the ecological settings of rice. This has already been published in Wood and Lenné (2018) and Lenné and Wood (2022). This does not exclude other environmental stresses if these are important to other agricultural settings.

L 115-116, Again, it would be helpful to cite ecological studies comparing the vegetational and biological diversity in these monodominant systems observed in nature as compared to modern intensive agricultural monocultures.

This is outside the scope of this paper.

L 122-124, Awkward, please rephrase.

Revised

L 124, What is “species-diverse-species vegetation”? Please clarify.

Thank you – we have revised this as Reviewer 1 also pointed it out.

L 125-130, It is well understood that different pest species react differently to varied different vegetational diversification settings and that behavioral or dispersal models cannot be applied universally. Under different vegetational settings, such as under forestry, diversity may also have a differential impact on stability. However, the observed variability in response to diversity, and a lack of ‘universality’ don’t in themselves invalidate the theory that diversified systems may increase stability under some or most agroecological settings.

We do not query this statement and we are not trying to refute the existence of cases where diversity may increase stability in some agroecosystems in the face of pest attacks. We are citing Janzen in two papers to show his change of thinking about the existence of natural monocultures. We do not use the term “universality” in this context. The reviewer is challenging a quotation from Janzen (1974). 

L 146, Re: “In fact, these communities are among the most fragile.” No citations are provided to back up such a strong statement. For one example on the fragility of large-scale tree monocultures, based on a large-scale multi-year experiment, please refer to the experience of the Great Green wall of China (Three-North Shelter Forest Program).

The statement on L146 is cited as Dover and Talbot (1987) as is clear in L140. They observed that if true, it provides a strong practical argument for conservation – for maintaining diversity in ecosystems whenever possible as a natural buffer against perturbation. But as they said if the theory was flawed, basing policies on it could have unexpected and undesirable environmental consequences. For instance, if diversity causes stability, the most species-rich communities – such as tropical rain forests and coral reefs – should be able to withstand the greatest disruption at human hands. In fact, these communities are among the most fragile.

There is a substantial literature on the fragility of coral reefs and tropical rain forests. Based on the recent serious damage done by pollution and climate change to coral reefs through bleaching such as the Great Barrier Reef in Australia and the serious damage being done to the Amazon rain forest through drought, logging and mining, it is strange that the reviewer does not appear to know about this literature on fragility. Three references have been added for completeness.

L 146-148, How can researchers “conclude an idea”? And how does an “idea” and a “notion” prove a scientific hypothesis? The authors seem to be conflating experimental results with the personal opinion of some scientists, based on “notions,” “ideas” and on a selective and narrow analysis of the scientific literature.

Sentence revised in text. Dover and Talbot (1987) concluded that experimental evidence and theoretical analysis questioned the belief that diversity causes stability. It was oversimplified.

L 149-150, It would be helpful, for the reader, to provide an ecological characterization of a monodominant system. Other than the >60% of the monodominant plant species (percent ground cover?), how many other species are found in these systems, and how would this compare to the plant vegetational and germplasm uniformity observed under industrial monoculture systems? For instance, in natural grassland prairies in the USA over 100 plant species may be found in small areas (<2 Ha), and over 250 species in large areas (of 450 Hectares) (Risser, P.G., 1988. Diversity in and among grasslands (pp. 176-180). In: Biodiversity. Washington, DC: National Academy of Sciences).

The reviewer is questioning an accepted definition of monodominance (now defined in Box 1) – again referring to it as a “monodominant system” – we could not find a definition of “monodominant system”. We do not refer to monodominant systems anywhere in the paper. The rest of this comment is outside the scope of the paper.

L 169-170, The authors need to substantiate the statement that traditional paddy rice cultivation conducted over hundreds of years, following natural methods of production, is equivalent to green revolution production practices including the reliance on genetically uniform cultivars (narrow germplasm base), and the application of high rates of fossil fuel-based chemical pesticides and fertilizers. The authors also need to substantiate the idea that the genetic background or genetic pool of modern production systems (based on planting one cultivar over extended areas) is equivalent to the genetic background of traditional or indigenous production systems. While modern production systems rely on a very narrow genetic pool, traditional farming communities, within the same geographical area (e.g. Asia, Africa, Latin America), often relied on a large number of varieties and landraces, representing a wider genetic pool.

Reference to the Green Revolution in this context has been removed which makes the rest of the reviewer’s comments redundant. We do not need then to substantiate any idea – which in fact we did not put forward – that the genetic background of modern rice cultivars is equivalent to the genetic background of traditional rice farming systems. More so, there is overwhelming evidence that the genetic diversity of current improved rice varieties is functionally more diverse than what was grown previously in traditional farming systems. For example, the popular variety IR64 which is widely grown over 10 million has in SE Asia has a pedigree of  79 parents including 20 landraces, 58 older varieties and one wild species Oryza nivara (Mackill and Khush, 2018). This functional diversity has been sourced from a number of countries. I would challenge the reviewer to locate any traditional farming community which is growing landraces with this level of functional diversity.

L 186-187, The authors are making a false-equivalence comparison, by comparing two separate systems that are unrelated to each other. By stating that Monodominance systems occur in some instances in nature (systems that the authors have failed to properly characterize in terms of their vegetational diversity, and percent area coverage in relation to all terrestrial ecosystems), the authors claim that industrial monoculture systems are also ‘stable’ which are contradicted by an extensive scientific record that has documented the instability of agricultural monocultures (such as their susceptibility to pest outbreaks, climatic events and broken nutrient cycles).

We have rewritten this section based on comments by reviewer 1. Nowhere in this paper do we use the term “industrial monoculture” in fact we do not qualify type of crop monoculture with any adjectives at all except in the definitions section to illustrate the problems with the interpretation on what a monoculture is. We never make the claim that “industrial monoculture systems are stable”. We have already answered the request for information on monodominance in nature with many examples in the paper as well as further references with even more examples.

L 220-223, Here, to support their claim, the authors should cite specific examples in which breeders have developed pest resistant varieties with anti-feeding compounds. The fact that crop breeders aim to develop resistant varieties does not support the authors’ claim, that resistance develops naturally under monocultures, especially with varieties that have a uniform genetic background— as is common in commercial crop monocultures.

We have revised the text to modify the apparent claim and added relevant references.

We do not claim that “resistance develops naturally under monocultures with uniform genetic background” or in “commercial crop monocultures”. The reviewer should be aware that “uniform genetic background in a stand of one crop variety masks considerable functional diversity as explained above for the rice variety IR64. We have indicated that anti-insect feeding compounds exist in a number of plants that grow in monocultures which must have evolved under insect pressure – that is how evolution works. This provides a basis for exploring crop gene pools for similar resistances. 

During pre-domestication it is probable that genetically variable monodominant stands of the wild ancestors of crops evolved resistances to prevailing pests as they could not have survived without doing so. Domestication resulted in bottlenecks as not all variation moved into the crops and useful characters were left behind. These are now the target of crop breeders. Hence formal crop breeding for resistance to insect pests provides on-going support for reducing damage by pests.

The reviewer continues to refer to the “genetically uniform” nature of improved varieties in current crop monocultures. What is ignored is the substantial genetic diversity for functional traits in these varieties. For example as noted above, the popular rice variety IR64 which is widely grown throughout SE Asia is substantially diverse with 79 parents including 20 landraces, 58 older varieties and one wild species Oryza nivara in its pedigree (Mackill and Khush, 2018).

L 224, In this section the authors need to explain why extensive tillage and pesticide applications are still required for the production of introduced cereal crops (as compared to the center of origin), even though they have reached ‘Monodominance’ by not having to confront the presence of “coevolved pests and diseases”. The authors also need to explain the frequent occurrence of crop epidemics, such as the potato leaf blight epidemic in Ireland during the 1800s, even though these epidemics have occurred in “Introduced areas.”

The first request is outside of the scope of the paper – we are focused on pest management not production e.g. tillage. The need to use pesticides in certain circumstances is also outside the scope. We dispute the claim of “frequent occurrence of crop epidemics” and it is interesting that the reviewer has only cited an epidemic of potato blight which occurred more than 180 years ago. We have provided the explanation of this very rare example of a pathogen which did not evolve with its host in the Andes but newly encountered it in Europe when it was introduced from Mexico via the USA (see above). Crop introduction away from indigenous pests and diseases is not fail safe We have acknowledged the importance of quarantine which greatly reduces the risk of pest and pathogen introductions but these can still occur.

L 256-257, This sentence is unclear, poor syntax. Also, provide citations, such as for cereals, root crops, and indigenous leafy crops-- to back up the statement.

We revised the sentence. We are talking about staple food crops in Africa: maize, cassava, sweet potato, Asian rice, East African highland banana – common staple food crops are all introduced to the African continent.

L 258-271, The authors here, again, have to reconcile the fact that a considerable amount of external inputs, adapted tillage, and pesticides are required to grow soybeans in Latin America even though pests and diseases have been “left behind an oceanic barrier.” To back-up their claims about Monodominance, the authors should provide comparative data, with citations, on the number of major pests and the levels of soybean crop losses both from China (center of origin) and Latin America (introduced species) when grown under monoculture and WITHOUT pesticide applications.

This is outside the scope of this paper. Export volumes alone show that soybean can be successfully grown and exported in comparison to much lower production in China. Again the reviewer returns to pesticide use as he/she considers pesticides to be inherent to monoculture production which we have already indicated several times is not the case. However the literature indicates that there are 20 key pests of soybean in China and only one – stink bug – in Brazil but the details are outside the scope of this paper.

274-276, Poor sentence structure.

Revised

272-280, With their claim that indigenous crops are “disadapted,” the authors need to reconcile the apparent contradictions, for example, with the successful production of indigenous crops such as rice in China, potatoes in the Andes, corn in the Americas, and root crops in Asia and the Pacific-- over hundreds of years, without the need of external chemical inputs— in contrast to the chemical intensive systems observed today for the production of introduced species in countries such as the USA (or soybeans in Latin America).

Revised – the paper is not about intensive or otherwise pesticide use as emphasized by the reviewer. There are many examples of introduced crops growing very well without pesticides in many countries e.g. rubber in SE Asia, tea in Sri Lanka, coffee in Brazil and Vietnam etc. Again, extensive crop breeding supports these crops where necessary and quarantine regulations further protect them.

L 224-319, The purpose of section 5 is unclear. Yes, initially, after introduction, crops may be grown without many of the major pests found in their center of origin. However, as indicated in L 302, eventually sporadic pest outbreaks become the norm with introduced crops, due in part to their vulnerability when grown under monoculture systems. Major pest epidemics have been experienced with introduced crops in the USA, such as with corn, wheat, and many vegetable species. Monodominance is then maintained through the reliance on high rates of external chemical inputs.

This section has been substantially revised. As I said before corn/maize is NOT an introduced crop – it is native to the Americas. We have indicated with examples that pests may catch up decades later with introduced crops however the widespread use of resistance breeding largely prevents major pest epidemics. The only ones that the reviewer mentions are rust of wheat in an earlier comment. The reliance on the use of chemicals is an on-going criticism specially the emphasis on high rates. I would suspect that USA and other country food safety regulations would have a role in restricting the rates used. Also, millions of farmers in less developed countries grow monocultures without pesticides as they cannot afford them. They are supported by crop breeding for resistance.

L 320-354, In this section the authors cite only one study published over the past 5 years. The authors support their case with respect to the effect of vegetational diversification, based on undefined and non-specific generalizations, such as using the terms “inconsistent” or claiming that “it is not necessarily true.” It would be more valuable to report the total number of published studies that have shown lower pest levels due to diversification vs. the number of studies that have shown no effect on diversification. It is well known that some pests species are better managed under diversified systems, than other species (due to variables such as their host range and dispersal behavior)— and thus it would be misleading to make blanket assessments without characterizing the type of pests, under discussion.

This section specifically cites reviews/meta-analyses (published from 2022-2023) which in several cases review over 100 studies – in total over 300 studies were analysed. All of the studies concluded that the findings on the use of diversity to manage pests generated mixed or inconsistent results.  These are neither blanket nor misleading as we are citing other authors who provided the data to support the results. Logically, the section concludes with highlighting the lack of research which was noted by most of the authors cited. In asking for a report of the total number of studies on this topic, the reviewer is asking for a far more detailed and comprehensive paper which is beyond the scope of this review.

L 384, It would be helpful for the authors to characterize what they mean by rice “monoculture” based on the region of production. For example, in Asia, the authors should describe the size of the plots in small vs large farms, the incidence and type of vegetation in the field borders, the incidence of rotations with other crops, as well as the vegetational diversification from a landscape perspective (such as the ‘mosaic’ cropping patterns as described in L 405)— and compare these systems with the larger scale monocultures found under industrialized systems, such as in the USA.

We have defined monoculture in Box 1 and provided further explanation in the sub-section under the Box. The definition of a monoculture does not include a defined area just a given area – it is a monospecific stand in a given area which may be 10 x 10 m or 100 x 100 m square. Size is not part of the definition of a monoculture. Irrigated monoculture rice - whether in small plots in SE Asia or as large-scale rice production in other parts of the world – is still a monoculture based on the original definition of the term. The reviewer is again confusing how the crop is managed with the definition of a monoculture.

L 401-404, This statement uses a very strong statement from a scientific perspective (“conclusively”), by citing two studies published over 25 years ago. To back up their statement, the authors again use ambiguous language, i.e. “not major factors,” which leaves the reader with the question of what is exactly meant by “not major factors” (only 10, 20 or 48% contribution to natural enemy activity?).

The reviewer is questioning already published studies which show these results. We have removed the word “conclusively”. The reviewer can read the Karp et al 2018 meta-analysis which has over 150 authors for more information in the studies.

L 414-415, Again, it is ambiguous or unclear to define the rice systems in Asia as monocultures, if the authors have failed to appropriately characterize the systems, including the level of biological diversity within and around the paddies (such as species of algae, azolla, aquatic life, and non-crop vegetation within and around the production fields). Why not provide comparative data with rice monocultures in the USA, as well as its production under dry land or upland conditions, rather than selectively focusing on only one type of production system?

Irrigated rice systems in Asia, USA and elsewhere are monocultures based on the original and accepted definition of the word. Also we are focussing on irrigated rice not other forms of management. The definition of monoculture does not include size – a monoculture is a monospecific stand. It also does not include other biodiversity that may be part of this system such as aquatic life and non-crop vegetation on bunds etc. We are not challenging diversity of other organisms in the irrigated rice crop system– monoculture is a specific and defined term. The reviewer is confused about what a monoculture is.

L 436, It is misleading for the authors to claim that, according to ¬¬¬-proponents, the buildup of natural enemy activity via crop diversification should be sufficient to control important pests “without the need for further intervention.” It is well understood by pest management specialists that a multiple number of management tactics are required as part of an integrated pest management program, and that it is ineffective or shortsighted to rely on only a single management practice. Thus, the use of crop diversification to build up the activity of beneficial organisms, or to deter pest infestations, would be considered only one aspect of the overall pest management program, along with improved soil quality, crop sanitation, organic mulches, resistant varieties, among other cultural management strategies.

We have removed “without the need for further intervention”

L 453, References, The authors should follow Plants-MDPI literature citation guidelines.

All references have been checked

Additional edit suggestions/questions are included in the attached copy of the manuscript.

Reviewer 3 Report

Comments and Suggestions for Authors

This Reviewer seas the need for discussion of this important topic. There is no question that what the authors term monodominance occurs in natural ecosystems, even if the factors leading to its occurrence may vary with context and might not be fully understood.  However, the logic is weak and the evidence selective in many areas of the paper.

1.      There is also no question that many if not most (but not all) managed agroecosystems that have existed for centuries or millennia are highly diverse. There are many reasons for such diversity. The reviewer agrees with those researchers (that the authors criticise) in proposing that diversity brings resilience to many external stresses. This is not to say that all low diversity systems are vulnerable, but the transition in many parts of the world to large areas of near monoculture, in both cereals and the main pasture grasses, has become a major challenge to sustainable agriculture.

2.      This reviewer’s principle criticism, however, is due to the focus on crop protection against pests (used in its widest sense) when plant diversity also has many other functions. For example, in traditional or indigenous agriculture, people have needed to get most of their food and other natural products from a defined and usually small area of land. Therefore they have to grow several different crops in close proximity to each other. A further reason is that different crops undoubtedly complement each other – for example N-fixing legumes increase the N availability to other crops. These wider arguments seem to be ignored in the present paper with its emphasis on insect pest control.

3.      In revision, the paper should consider and discuss that the degree of crop diversity has far more influences than on pest regulation. Also it should consider that industrialised crop monocultures are causing a range of problems in many parts of the world, including increased erosion, destruction of farmland biodiversity, the need for large amounts of industrially manufactured fertiliser, and rising pesticide usage (see below).  

4.      A substantial – and surprising - omission from the argument is the rise in pesticide usage in industrialised monocultures. The authors should present detail of the rise on pesticide application globally and evidence of how pesticides might be reduced. Part of the problem is that pests evolve resistance to pesticides, which generally leads to an increase in applications or the introduction of new chemicals. A full discussion of this topics has to be included.

5.      The arguments around ‘flight response’ especially for weeds (in part 5 for example) need to be seriously reconsidered. Yes, crops probably moved between continents faster than their associated weeds, but they did not escape pest issues as the authors appear to be suggesting. In almost every part of the world where crops have been introduced, weeds (and other pests) take advantage and cause problems for crop production. If the authors wish to promote the idea that trans-continental movement allows crops to evade pests, then they need to provide hard evidence.

6.      Following point 5, industrialised ‘monocultures’ are rarely monocultures. If weeds are not stringently controlled, then they will come to coexist with crops. Many historical records, and much current experience in low-input agriculture, in fact show how serious was is this problem of monocultured crops (from other continents) transitioning to polycultures of crops and weeds. Most industrialised crops are near-monocultures only because of excessive tillage and agritoxin usage to remove the other plants.

7.      In sections 4 and 5, the logical flow of the argument tends to get lost in a desire to criticise other researchers. (In contrast, the arguments are stronger in part 6.) For example, it is difficult to appreciate what sentences like this mean: “It is tempting to relate the increasing dominance of South America in soybean production and export with the campaign promoting mixed cropping, as advised by Tilman from North America and, most notably, the striking over-promotion of agroecology for Latin America in the World Bank’s International Assessment of Agricultural Science and Technology for Development 270 (IAASTD) regional report (MacIntyre et al., 2009).”

8.      This reviewer recommends that the authors take a critical look at the logic and presentation of evidence they present, especially in parts 4 and 5, and particularly with regard to comments above.

Comments on the Quality of English Language

No specific comments

Author Response

Reviewer 3

This Reviewer sees the need for discussion of this important topic. There is no question that what the authors term monodominance occurs in natural ecosystems, even if the factors leading to its occurrence may vary with context and might not be fully understood.  However, the logic is weak and the evidence selective in many areas of the paper.

  1. There is also no question that many if not most (but not all) managed agroecosystems that have existed for centuries or millennia are highly diverse.

We are not arguing with the existence of diversity in agroecosystems but we disagree that most crop fields are diverse. Crop-livestock farms are one of the best widespread examples of diverse farms. Also there are good examples in remnant traditional systems such as the milpa system in Central America. However most crop fields are farmed as monocultures as they are easy to plant, manage and harvest.

There is irrefutable evidence – much of it cited in Wood and Lenné (2018) that pre-domestication the wild relatives of the founder cereals – wheat, barley and rice were gathered from monocultures.

There are many reasons for such diversity. The reviewer agrees with those researchers (that the authors criticise) in proposing that diversity brings resilience to many external stresses. This is not to say that all low diversity systems are vulnerable, but the transition in many parts of the world to large areas of near monoculture, in both cereals and the main pasture grasses, has become a major challenge to sustainable agriculture.

If this is so then why is most of our staple food from monoculture agriculture – as it has been for millennia and currently feeds more than 8 billion people. Yes, we agree that improvements can be made and need to be made in how they are managed but we challenge that monocultures are a major challenge to sustainable agriculture.

  1. This reviewer’s principle criticism, however, is due to the focus on crop protection against pests (used in its widest sense) when plant diversity also has many other functions. For example, in traditional or indigenous agriculture, people have needed to get most of their food and other natural products from a defined and usually small area of land. Therefore they have to grow several different crops in close proximity to each other. A further reason is that different crops undoubtedly complement each other – for example N-fixing legumes increase the N availability to other crops. These wider arguments seem to be ignored in the present paper with its emphasis on insect pest control.

We were asked to write a paper focused on pest management. We did not purposely ignore the other benefits of diverse agroecosystems but the paper is for a special issue on pest management.

  1. In revision, the paper should consider and discuss that the degree of crop diversity has far more influences than on pest regulation. Also it should consider that industrialised crop monocultures are causing a range of problems in many parts of the world, including increased erosion, destruction of farmland biodiversity, the need for large amounts of industrially manufactured fertiliser, and rising pesticide usage (see below).

As above we were asked to write the paper in the context of pest management. We did not define any specific type of monoculture – there are many different types of monocultures depending on how they are managed. This includes so called industrial crop monocultures but there are also many others that are not “industrial” and are grown throughout the world.

  1. A substantial – and surprising - omission from the argument is the rise in pesticide usage in industrialised monocultures. The authors should present detail of the rise on pesticide application globally and evidence of how pesticides might be reduced. Part of the problem is that pests evolve resistance to pesticides, which generally leads to an increase in applications or the introduction of new chemicals. A full discussion of this topics has to be included.

The paper is not about industrial monocultures or about pesticide use – it is about crop and agroecosystem diversity and pest management. I am sorry but this is outside the scope of this paper. You have not mentioned a role for host plant resistance which has made a major contribution to pest management in monocultures. We have highlighted this.

  1. The arguments around ‘flight response’ especially for weeds (in part 5 for example) need to be seriously reconsidered. Yes, crops probably moved between continents faster than their associated weeds, but they did not escape pest issues as the authors appear to be suggesting. In almost every part of the world where crops have been introduced, weeds (and other pests) take advantage and cause problems for crop production. If the authors wish to promote the idea that trans-continental movement allows crops to evade pests, then they need to provide hard evidence.

We have indicated in the section on definitions that we are focusing on insect pests and diseases and not on weeds. Weeds could be considered in another paper.

  1. Following point 5, industrialised ‘monocultures’ are rarely monocultures. If weeds are not stringently controlled, then they will come to coexist with crops. Many historical records, and much current experience in low-input agriculture, in fact show how serious was is this problem of monocultured crops (from other continents) transitioning to polycultures of crops and weeds. Most industrialised crops are near-monocultures only because of excessive tillage and agritoxin usage to remove the other plants.

We are not talking about weeds or industrialised monocultures in this paper.

  1. In sections 4 and 5, the logical flow of the argument tends to get lost in a desire to criticise other researchers. (In contrast, the arguments are stronger in part 6.) For example, it is difficult to appreciate what sentences like this mean: “It is tempting to relate the increasing dominance of South America in soybean production and export with the campaign promoting mixed cropping, as advised by Tilman from North America and, most notably, the striking over-promotion of agroecology for Latin America in the World Bank’s International Assessment of Agricultural Science and Technology for Development 270 (IAASTD) regional report (MacIntyre et al., 2009).”

Much of the criticism of other researchers has been removed as was also requested by the other reviewers of the paper. We have slightly modified the reference to the IAASTD. The context of the IAASTD and its promotion of diversity-based agroecology versus monocultures is however important to mention in this perspective paper.

  1. This reviewer recommends that the authors take a critical look at the logic and presentation of evidence they present, especially in parts 4 and 5, and particularly with regard to comments above.

Modifications have been made to the paper based on all reviewers’ comments. We feel that the presentation is now logically flowing from one section to another.

Round 2

Reviewer 1 Report

Comments and Suggestions for Authors

The authors have made in depth modifications to address the previous comments. I am completely satisfied with them. I congratulate them on their informative and provocative contribution.

Author Response

The authors have made in depth modifications to address the previous comments. I am completely satisfied with them. I congratulate them on their informative and provocative contribution.

The authors thank reviewer 1 for the comments.

Reviewer 2 Report

Comments and Suggestions for Authors

The authors have streamlined an earlier version of the manuscript, and also kindly responded to each of the questions and suggestions made by the external reviewers. The manuscript now reads and flows better and includes a helpful Table with definitions (however some modifications are suggested, see below).

While this paper is a "Perspective" it is important that the authors support their hypothesis with a rigorous scientific background. While monodominance has been recorded in some ecosystems, it may still be unclear to the reader how does a natural ecoystem system of monodominance compares (say in tropical wetlands, or semi-desert/arid regions) with today's highly industrialized monoculture systems (covering often millions of acres, with a relatively uniform genetic base), i.e. is the analysis valid or is it comparing "apples" with "oranges." To better compare both systems, it would be helpful for the authors to characterize a monodominant system for mediterranean grains, as suggested in my original review. While the authors may have cited some of their early work on monodominance, it would be helpful for the reader, to do so in this manuscript, as well (it may take a few sentences to do so). It is important that the authors provide a rigorous scientific analysis, to support their hypothesis, which counters the prevailing scientific consensus which is critical of habitat simplification, and in support of diversified systems (Rasmussen et al., Joint environmental and social benefits from diversified agriculture, Science Mag. 384,87 (April 5, 2024)).

It would also be helpful, for the reader, to describe to what type of modern agricultural systems does the "monodominant" hypothesis pertains to. Does it apply solely to mediterranean cereals and rice, or does it apply to all agricultural crops in general? How come maize, a major global cereal, is not considered in the analysis?

Additional comments on the manuscript include,

L 57, Suggest to include citations in the table, for each of the definitions.

L 57, As part of the definition of Monodominance, please clarify if under natural systems, under monodominance, the plant species are genetically uniform, or if the population consists of plants with a wide genetic background (such as a mixed population of wild relatives, and land races).

L 57, The definition of “diverse agroecosystems” is unclear. How can agricultural fields be considered “natural systems,”? i.e. how can human modified landscapes be considered “natural systems”?

L 59-62, In this sentence the authors are providing their opinion, which is supported in the text by citing the opinion of other authors. While this paper is a ‘perspective’ the information provided should still be based on rigorous scientific analysis. A scientific hypothesis, can’t be based nor proven by unsupported personal opinions.

L 65-69, The authors are conflating the terminology used in the manuscript, which lacks precision. Loose or unclear definitions makes it difficult to establish a solid foundation, in support of an hypothesis. The authors seem to be trying to provide a rather broad definition of monocultures, which goes beyond what is generally understood as “monocultures” in the standard agriculture literature (for example, referred to in the literature as habitat “simplification, for example, in the form of intensively managed monocultures” Rasmussen et al., 2024). It is not reasonable for the authors to selectively broaden their definition of ‘monocultures’ with the goal of supporting their hypothesis on monodominance. Monocultures, in the conventional sense, are characterized by large expansive production areas (not 10 x 10 meter plots), for the production of crops with a highly narrow genetic base (in which the authors are again conflating the terminology, by referring to this narrow genetic base, as “functional genetic diversity”). There is an international scientific consensus that modern agriculture is experiencing a dangerous decades long trend of genetic “erosion” esp. given the recent consolidation of the seed Industry. The planting of a mixture of varieties and land races on the same field could probably better be described as polycultures or as traditional or indigenous production systems.

L 187-188, Who has disputed the existence of monodominant vegetation in natural systems? However monodominant ecosystems need to be properly characterized, in terms of the respective climatic conditions, trophic level complexity, and vegetational diversity i.e. actual number of plant species in a given area/ecosystem.

L 302-306, This sentence is unclear. What is the point?

L 311-314, The authors need to reconcile this statement with the emergence of a number of pests in this species. For example a recent review on Pinus indicates that “It is affected by a wide range of plant-feeding insects both in its native range and in regions where it is planted as an introduced tree” (Brockerhoff, et al, 2023. NeoBiota, 84, pp.137-167.),

L 322-332, What is the point of these sentences? How does it support the paper’s hypothesis? But again, with respect to this section, what is the point of monodominance and the “escape response”, when invasive and new pests are a common occurrence with all major crops around the world, regardless of their center of origin, leading to frequent epidemics?

Additional minor edit suggestions are included in the attached copy of the manuscript.

/////

Comments on the Quality of English Language

The manuscript requires moderate editing.

Author Response

The authors have streamlined an earlier version of the manuscript, and also kindly responded to each of the questions and suggestions made by the external reviewers. The manuscript now reads and flows better and includes a helpful Table with definitions (however some modifications are suggested, see below).

While this paper is a "Perspective" it is important that the authors support their hypothesis with a rigorous scientific background. While monodominance has been recorded in some ecosystems, it may still be unclear to the reader how does a natural ecoystem system of monodominance compares (say in tropical wetlands, or semi-desert/arid regions) with today's highly industrialized monoculture systems (covering often millions of acres, with a relatively uniform genetic base), i.e. is the analysis valid or is it comparing "apples" with "oranges." To better compare both systems, it would be helpful for the authors to characterize a monodominant system for mediterranean grains, as suggested in my original review. While the authors may have cited some of their early work on monodominance, it would be helpful for the reader, to do so in this manuscript, as well (it may take a few sentences to do so). It is important that the authors provide a rigorous scientific analysis, to support their hypothesis, which counters the prevailing scientific consensus which is critical of habitat simplification, and in support of diversified systems (Rasmussen et al., Joint environmental and social benefits from diversified agriculture, Science Mag. 384,87 (April 5, 2024)).

Response: Unfortunately we continue to disagree with the request of this reviewer. We have already provided comprehensive reasons for this in the response to the comments on the first version of the paper. Monodominance has been recorded widely in many ecosystems. We have provided a number of our references which document this in the paper. For example:

Lenné J. Wood D. Monodominant natural vegetation provides models for nature-based cereal production. Outl. Agric. 2022, 51, 11-21. doi: 10.1177/00307270221078022.

Wood D. Lenné J. Nature’s fields: a neglected model for increasing food production. Outl. Agric. 2001 30, 161-170. doi.org/10.5367/000000001101293616

Wood D. Lenné, J. M. A natural adaptive syndrome as a model for the origins of cereal agriculture’, Proc. R. Soc. B 2018, 285, 20180277. doi: http://dx.doi.org/10.1098/rspb.2018.0277.

Again the reviewer refers only to highly industrialized monocultures covering millions of hectares with a relatively uniform genetic base – we have already shown that monocultures come in all shapes and sizes and are grown by small, medium and large farmers globally using many kinds of management methods (L59-69). We have also challenged the reviewer regarding the use of the term “uniform genetic base”. I have already given the reviewer the example of the popular rice variety IR64 which is widely grown over 10 million has in SE Asia has a pedigree of  79 parents including 20 landraces, 58 older varieties and one wild species Oryza nivara (Mackill and Khush, 2018). This functional diversity has been sourced from a number of countries. Yes, the variety is genetically uniform but intra-varietally extremely diverse. Would the alternative be to mix up all 79 parental lines and sow those to meet the requirement for genetic diversity? Morphological uniformity in modern improved crop varieties hides a great deal of genetic diversity from many sources and is unfortunately little understood outside of plant breeders. The analysis called for by the reviewer is meaningless.

I am not sure what the “Mediterranean grains” is about. We refer only to the fertile crescent and not to the Mediterranean. The origins of wheat and barley are firmly anchored in the fertile crescent.

We have already added a paragraph from the paper (L 110-123):

Wood D. Lenné, J. M. A natural adaptive syndrome as a model for the origins of cereal agriculture’, Proc. R. Soc. B 2018, 285, 20180277. doi: http://dx.doi.org/10.1098/rspb.2018.0277

This documents much of the early work on the monodominance of the founder cereals.

The reviewer uses a re-defined version of “monoculture”; we prefer to use the original definition for clarity. 

In this paper, we question the apparent prevailing scientific consensus critical of habitat simplification and supporting diversification. We read the Rasmussen et al paper. It starts from the premise that simplification of agriculture is based in industrialization of agriculture which is not correct. Where is the evidence that agriculture began in diverse systems? In fact the available archeological evidence at least for cereals supports simplified systems. For example at Ohalo II in northern Israel there are cereal deposits dating back 23,000 years showing that gatherers harvested seed from near monocultures of cereal wild relatives. Early farmers presumably copied the existing wild stands with domesticated cereals from 11,000 years ago – again as simple systems.

 In addition, the paper is not only about pest management but also about food production. Currently mainly monoculture agriculture feeds 8 billion people. Where is the data to show that diversified systems can do the same?

It would also be helpful, for the reader, to describe to what type of modern agricultural systems does the "monodominant" hypothesis pertains to. Does it apply solely to mediterranean cereals and rice, or does it apply to all agricultural crops in general? How come maize, a major global cereal, is not considered in the analysis?

Response: We have indicated that the hypothesis is based on the founder cereals – wheat, barley and rice. We have suggested that it probably applies to other cereals whose wild relatives had large seed and awns (L 136-138). We have also noted that further research is needed on the ecology and domestication of other crop groups. I acknowledge that maize is a mystery. The genetic mutation that caused teosinte to form maize must have been massive. It is a pity that not more research was done on the wild relatives of maize before they were reduced to remnant populations.

Additional comments on the manuscript include,

L 57, Suggest to include citations in the table, for each of the definitions.

  We have added sources of definitions in the box as notes.

L 57, As part of the definition of Monodominance, please clarify if under natural systems, under monodominance, the plant species are genetically uniform, or if the population consists of plants with a wide genetic background (such as a mixed population of wild relatives, and land races).

Response: We have included the accepted definition of monodominance – 60% or more of plants of the same species. The “same” species does not preclude intra-specific variation – whether in wild relatives or landraces which we have added. But this is not included in the definition.

L 57, The definition of “diverse agroecosystems” is unclear. How can agricultural fields be considered “natural systems,”? i.e. how can human modified landscapes be considered “natural systems”?

The definition has been clarified.

L 59-62, In this sentence the authors are providing their opinion, which is supported in the text by citing the opinion of other authors. While this paper is a ‘perspective’ the information provided should still be based on rigorous scientific analysis. A scientific hypothesis, can’t be based nor proven by unsupported personal opinions.

Response: We have added several notes to support this statement which is overwhelmingly true and in fact reflected in this reviewer’s bias against monoculture/simplified agriculture. The Sumberg and Giller reference provides more support for the statement.

L 65-69, The authors are conflating the terminology used in the manuscript, which lacks precision. Loose or unclear definitions makes it difficult to establish a solid foundation, in support of an hypothesis. The authors seem to be trying to provide a rather broad definition of monocultures, which goes beyond what is generally understood as “monocultures” in the standard agriculture literature (for example, referred to in the literature as habitat “simplification, for example, in the form of intensively managed monocultures” Rasmussen et al., 2024). It is not reasonable for the authors to selectively broaden their definition of ‘monocultures’ with the goal of supporting their hypothesis on monodominance. Monocultures, in the conventional sense, are characterized by large expansive production areas (not 10 x 10 meter plots), for the production of crops with a highly narrow genetic base (in which the authors are again conflating the terminology, by referring to this narrow genetic base, as “functional genetic diversity”). There is an international scientific consensus that modern agriculture is experiencing a dangerous decades long trend of genetic “erosion” esp. given the recent consolidation of the seed Industry. The planting of a mixture of varieties and land races on the same field could probably better be described as polycultures or as traditional or indigenous production systems.

Response: As I said above this is biased criticism – highlighted in red. We have defined “monoculture” according to the accepted definition – the cultivation of a single crop - a monospecific stand – in a given area. It is not loose or unclear. It is the accepted definition by many well-regarded dictionaries and encyclopedia. We have not broadened the definition. What is generally understood by the reviewer is not a definition. If large expanses of production are now generally understood then I suggest that those that understand this should revisit the original definition or visit a developing country. A monoculture can be 10m x 10m which is common in smallholder fields in developing countries. Together we have over 60 year’s experience working in developing countries in Africa, Asia and Latin America and have seen thousands of monocultures of this type. The reviewer seems to think that the only monocultures which exist are large and industrial.  

Reference to the “decades long trend of genetic erosion” and the consolidation of the seed industry have nothing to do with this paper. Also I have strongly challenged the basis of the “decades long trend in genetic erosion” in another paper:

Lenné, Jillian (2023) Current agricultural diversification strategies are already agroecological. Outlook on Agriculture 52: 273-280.

According to FAO (1993) ‘Since the 1900s, some 75 percent of plant genetic diversity has been lost as farmers worldwide have left their multiple local varieties and landraces for genetically uniform, high-yielding varieties’. However Brush (1995) noted that 20 years after the genetic erosion alarm was raised, the extent of erosion has not been measured quantitatively and Ceccarelli et al.

(1992) have pointed out that, little is known about the actual rate of genetic erosion of crops and their wild progenitors. In a forensic search of FAO documentation, Khoury et al. (2022) traced this 75% figure back to an estimate of loss of three quarters of vegetable varieties in Europe by the Canadian Rural Advancement Fund International (RAFI) (Fowler and Mooney, 1990). This morphed into 75% loss of all crop genetic diversity in FAO (1993), also written by RAFI, and has become FAO-lore. Furthermore, it must be emphasized that loss of varieties cannot be equated with loss of diversity, as new varieties are derived from old varieties with similar genetic diversity. Khoury et al. (2022) noted that the 75% narrative was a powerful tool to raise awareness, yet one must question whether using an unproven claim to pressure farmers to grow more diversity is ethical.

L 187-188, Who has disputed the existence of monodominant vegetation in natural systems? However monodominant ecosystems need to be properly characterized, in terms of the respective climatic conditions, trophic level complexity, and vegetational diversity i.e. actual number of plant species in a given area/ecosystem.

Modified

L 302-306, This sentence is unclear. What is the point?

Modified

L 311-314, The authors need to reconcile this statement with the emergence of a number of pests in this species. For example a recent review on Pinus indicates that “It is affected by a wide range of plant-feeding insects both in its native range and in regions where it is planted as an introduced tree” (Brockerhoff, et al, 2023. NeoBiota, 84, pp.137-167.),

Response: We have modified this sentence. The Brockerhoff paper is mainly about recordings of pest distribution and potential risks. In addition there are regions e.g. the Pacific where the tree has been introduced yet 86% of the native insects have not. This data is not provided for Australia and New Zealand where it is likely to be similar as it is said to be at risk. Both countries have excellent quarantine systems which to date appear to have successfully addressed the riskd. There is also no data on damage caused by non-native insects in this paper.

L 322-332, What is the point of these sentences? How does it support the paper’s hypothesis? But again, with respect to this section, what is the point of monodominance and the “escape response”, when invasive and new pests are a common occurrence with all major crops around the world, regardless of their center of origin, leading to frequent epidemics?

Response: We have indicated that introduction of crops to another continent away from their center of origin and native pests is not failsafe. Re-encounters can occur – fortunately not too often. The point of this sub-section is to show that scientific research on biocontrol has an important role in addressing such occurrences and can be very successful in the case of cassava mealybug in Africa.

Additional minor edit suggestions are included in the attached copy of the manuscript.

Minor edits have been addressed

Reviewer 3 Report

Comments and Suggestions for Authors

This reviewer notes the responses given by the authors, but feels that there are still major biases in the analysis and presentation. The authors are clearly knowledgeable of their topic, but prefer not to present data and enter into argument about some highly relevant issues such as the global increases in pesticide usage in intensified agriculture and the fact that many crop monocultures are not actual monocultures but assemblages of crops and weeds. While the authors assert that they were not asked to consider weeds, the reality is that weeds are important to insect pests and their predators and parasitoids.

Author Response

This reviewer notes the responses given by the authors, but feels that there are still major biases in the analysis and presentation. The authors are clearly knowledgeable of their topic, but prefer not to present data and enter into argument about some highly relevant issues such as the global increases in pesticide usage in intensified agriculture and the fact that many crop monocultures are not actual monocultures but assemblages of crops and weeds. While the authors assert that they were not asked to consider weeds, the reality is that weeds are important to insect pests and their predators and parasitoids.

Response: In any perspective paper, there will be biases – a perspective is an opinion piece which expresses an opinion supported by relevant literature. We did not aim to write a review presenting all views and relevant literature on the particular topic.

As we said, the definition of a monoculture is one of the most misunderstood and wrongly used terms in agriculture. As the definition box states: a monoculture is a single crop species - monospecific stand - on a given area. How it is managed is not a definition of monoculture. The use/overuse of pesticides in “industrial” agriculture is not relevant to this paper. We do not refer to management issues.

We agree that weeds can be a problem in agriculture – not just in monocultures. Weeds may host pests and/or predators and parasitoids – we present the example of irrigated rice with four decades of research to understand the pest management issues. We clearly state that the vegetation on the bunds – including weeds – is an important source of predatory spiders to control rice insect pests. In addition, as the accepted definition of monodominance is 60% or more of plants being of the same species, this would give allowance for weeds being part of a monoculture which may need to be controlled in some circumstances.  
